# Development and deployment of a histopathology-based deep learning algorithm for patient prescreening in a clinical trial

Albert Juan Ramon ⓘ[1] ✉, Chaitanya Parmar[1], Oscar M. Carrasco-Zevallos[2], Carlos Csiszer[3], Stephen S. F. Yip[2], Patricia Raciti ⓘ[4], Nicole L. Stone[4], Spyros Triantos[4], Michelle M. Quiroz[4], Patrick Crowley[5], Ashita S. Batavia[3], Joel Greshock[6], Tommaso Mansi[3] & Kristopher A. Standish[1]

Accurate identification of genetic alterations in tumors, such as Fibroblast Growth Factor Receptor, is crucial for treating with targeted therapies; however, molecular testing can delay patient care due to the time and tissue required. Successful development, validation, and deployment of an AI-based, biomarker-detection algorithm could reduce screening cost and accelerate patient recruitment. Here, we develop a deep-learning algorithm using >3000 H&E-stained whole slide images from patients with advanced urothelial cancers, optimized for high sensitivity to avoid ruling out trial-eligible patients. The algorithm is validated on a dataset of 350 patients, achieving an area under the curve of 0.75, specificity of 31.8% at 88.7% sensitivity, and projected 28.7% reduction in molecular testing. We successfully deploy the system in a non-interventional study comprising 89 global study clinical sites and demonstrate its potential to prioritize/deprioritize molecular testing resources and provide substantial cost savings in the drug development and clinical settings.

Comprehensive genetic and molecular testing of cancer tissue is critical for providing physicians with actionable insights to drive precise selection of targeted cancer treatments[1]. Currently, Fibroblast Growth Factor Receptor (*FGFR*) alteration testing is used to identify patients who could benefit from *FGFR*-targeted therapies such as BALVER-SA™(erdafitinib), the first targeted therapy approved by the FDA to treat patients with metastatic or locally advanced bladder cancer who have previously received platinum-based chemotherapy[2,3]. For instance, the QIAGEN Therascreen *FGFR* RT-PCR kit[4,5] is the companion diagnostic approved by the FDA to screen patients for the presence of specific alterations in *FGFR2* and *FGFR3* genes that determine eligibility for treatment with erdafitinib and is also used to identify which patients are eligible to enroll in clinical trials using erdafitinib to treat urothelial cancer[6–8].

While FGFR-targeted treatments improved clinical care[9], the wide adoption of molecular testing as standard of care remains slow due, in part, to its high cost and slow turn around, with an average 7 day turn-around time for test results[10–14]. Furthermore, molecular tests require substantial amounts of tissue with up to six, 4–5 μm sections of tumor tissue[15] and can fail to detect the target due to poor DNA/RNA quality

[1]Janssen R&D, LLC, a Johnson & Johnson Company. Data Science and Digital Health, San Diego, CA, USA. [2]Janssen R&D, LLC, a Johnson & Johnson Company. Data Science and Digital Health, Cambridge, MA, USA. [3]Janssen R&D, LLC, a Johnson & Johnson Company. Data Science and Digital Health, Titusville, NJ, USA. [4]Janssen R&D, LLC, a Johnson & Johnson Company. Oncology, Spring House, PA, USA. [5]Janssen R&D, LLC, a Johnson & Johnson Company. Global Development, High Wycombe, UK. [6]Janssen R&D, LLC, a Johnson & Johnson Company. Data Science and Digital Health, Spring House, PA, USA. ✉e-mail: ajuanram@its.jnj.com

and low tumor purity[16]. Moreover, *FGFR* genes are only mutated in 10–20% of advanced/metastatic urothelial cancer patients[16,17], resulting in a majority of test results coming back *FGFR-*. Therefore, finding fast, reliable strategies for screening patients is crucial for improving patient care and efficiently enrolling clinical trials.

Hematoxylin and Eosin (H&E) staining is a routine histopathological technique for diagnosing cancer that is affordable, widely practiced, and provides comprehensive visual representation of the tumor and associated microenvironment[18,19]. Previous work has demonstrated the feasibility of computational pathology algorithms for tumor classification and segmentation, mutation classification, molecular sub-typing, and outcome prediction from H&E images[20–32]. Furthermore, recent studies showed that some families of *FGFR* mutations may be associated with cytomorphologic tissue changes in H&E-stained images in urothelial carcinoma[33,34]. Loeffler[35] developed machine learning models to predict the *FGFR3* status of an individual's tumor on digitized histology slides[35–37]. Although promising, with an AUC of -0.7 on *FGFR3* mutations, these studies were performed on relatively small datasets (i.e., <300 whole slide images (WSIs)) and therefore their generalizability is not clearly understood. Furthermore, these algorithms did not focus on the currently clinically actionable class of *FGFR* alterations[4,5]. To our knowledge, there have not been any examples of these algorithms being used in a clinical trial setting. Deploying an H&E-based *FGFR+* screening device in clinical trials or clinical practice could be valuable in a number of ways: (i) reduce costs by avoiding molecular testing of patients that are unlikely to harbor genetic mutations, (ii) reduce the time to enrollment or access to the right targeted therapies by providing fast, actionable insight to physicians (i.e., enriching cohorts with patients likely being *FGFR+*)

In this work, we describe the development, validation, and deployment of a deep learning (DL)-based algorithm that infers the presence of specific *FGFR* alterations from common H&E-stained WSIs from patients with advanced urothelial cancers. Datasets from public repositories, commercial sources, and internal clinical trials are compiled and the algorithm is trained on H&E-stained WSIs from >3000 patients with urothelial carcinomas and *FGFR* mutation status. We perform a robust validation on multiple independent large-scale datasets and in a prospective real-time clinical setting following the international standard for device quality management systems (ISO 13485)[38]. The algorithm is deployed prospectively in a non-interventional clinical study[6] comprising 89 global study sites across 9 countries, to screen patients prior to molecular testing, enabling a physician to halt molecular testing for patients unlikely to harbor the targeted alterations and thereby saving tissue for other tests. We demonstrate this technology's potential to reduce screening burden

and improve trial efficiency. We believe this work also constitutes a step forward for precision medicine by enabling rapid, actionable clinical insight into a patient's specific disease and increasing access to effective, targeted therapies where approved for use.

## Results

### Algorithm development and packaging for deployment

To develop a robust algorithm for predicting *FGFR* genomic alterations from H&E slides, we used whole slide images from patients with muscle-invasive urothelial cancer (MIBC, pT2 or higher) or metastatic urothelial cancer from three different cohorts: 407 from The Cancer Genome Atlas (TCGA) consortium (https://portal.gdc.cancer.gov/projects/TCGA-BLCA), 2811 from BLC3001 (NCT03390504), and 184 from BLC2002 (NCT03473743), two erdafitinib trials[7,8]. The prevalence of *FGFR* in each cohort was 12.5%, 11.6% and 15.7% respectively, totaling a -12% average prevalence. Figure 1 shows the study design that was followed, which is explained further in the Methods sections. The Development Datasets were split into Training Data (85%, or 2820 slides) and Hold-out Data (15%, or 582 slides). The data split preserved the same ratio of *FGFR+* vs. *FGFR-* patients, as well as the proportion of samples from each cohort. The Training Data was used for algorithm optimization, and the Hold-out Data to assess algorithm performance prior to packaging it for the deployment platform. Once onboarded, a stepwise validation was performed, first with a well-powered Retrospective Validation to decide if the algorithm would be implemented in the clinical workflow, followed by a Deployment Setting Validation to assess workflow integration of the algorithm in the proposed deployment trial (ANNAR (NCT03955913)[6]).

In order to maintain high enrollment rates of *FGFR+* patients and to mitigate potential selection bias in trial participants upon deployment, the algorithm was required by the clinical and trial operations teams to achieve a target minimum sensitivity of 0.9 on Retrospective Validation. Thus, we performed hyperparameter tuning via 5-fold cross-validation and selected the best-performing model with a sensitivity above this target. Then, we evaluated its performance on: (1) the Hold-out Data (i.e., 582 slides from BLC3001, BLC2002 and TCGA); and (2) an independent dataset with slides from multiple solid tumor tissues (i.e., PAN-Tumor dataset with 361 slides).

Figure 2 shows the receiver-operating-characteristic (ROC) curves as well as the area under the ROC curve (AUC), area under the Precision-Recall curve (auPR), sensitivity and specificity values for each dataset. In the Hold-out Data (582 slides from BLC3001, BLC2002 and TCGA), the algorithm achieved an AUC of 0.80, auPR of 0.42, sensitivity 0.94 and specificity 0.38. We observed a slight drop in AUC (0.77), auPR (0.37) and sensitivity (0.92) but similar specificity (0.38) in

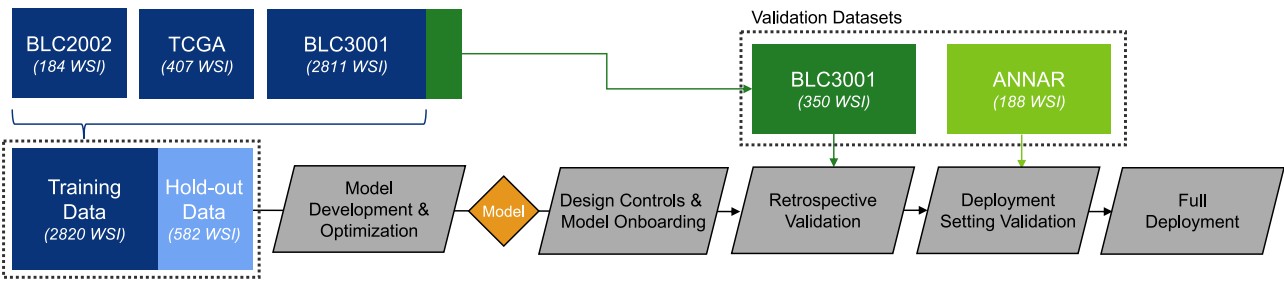

**Fig. 1 | Study design.** Dataset use and workflow from algorithm development through validation and deployment. Whole slide images (WSI) from three different cohorts were used for model development: 407 from The Cancer Genome Atlas (TCGA) consortium, 3161 from BLC3001 (NCT03390504) and 184 from BLC2002 (NCT03473743) from two erdafitinib trials[7,8]. A subset of 350 samples (150 *FGFR* +, 200 *FGFR*-; enriched for *FGFR+* to achieve a -93% statistical power) from the BLC3001 cohort, the trial with closest population to the deployment setting, and

188 samples from ANNAR (NCT03955913)[6], the deployment trial, were left out for Retrospective Validation after packaging the algorithm into a deployable device and onboarding on deployment platform. There were no patients used in both Development and Retrospective Validation. An additional cohort with 361 slides from multiple tumor tissues (i.e., PAN-Tumor) from a data vendor was used to evaluate performance of the algorithm on solid tumors as exploratory analysis after deployment of the tool.

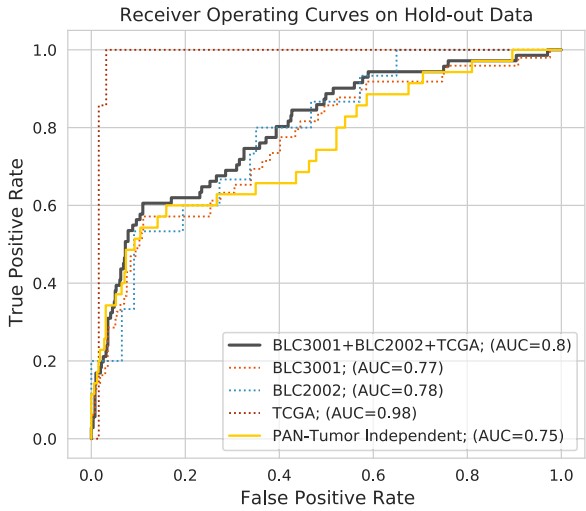

| | Hold-out Data | | | | |
|---|---|---|---|---|---|
| | BLC3001 BLC2002 TCGA | BLC3001 | BLC2002 | TCGA | PAN-Tumor |
| Dataset Size (FGFR+ prevalence) | 420+92+70 (~12%) | 420 (~12%) | 92 (~16%) | 70 (~10%) | 361 (~10%) |
| AUROC | 0.80 | 0.77 | 0.78 | 0.98 | 0.75 |
| AUPR | 0.42 | 0.37 | 0.49 | 0.68 | 0.39 |
| Sensitivity | 0.94 | 0.92 | 1 | 1 | 0.95 |
| Specificity | 0.38 | 0.38 | 0.28 | 0.48 | 0.27 |
| Reduction rate | 34% | 34% | 24% | 43% | 25% |

**Fig. 2 | Algorithm performance on held-out datasets.** In the left figure, shown in black is the performance on the Hold-out Data (582 slides from BLC3001, BLC2002 and TCGA); in red the performance on the subset with closest population to the one in the deployment setting (BLC3001 subset with 420 slides from the 582 slides) and in yellow the performance on an independent dataset with slides from multiple tumor tissues (i.e., PAN-Tumor with 361 slides). Performances are summarized in the legend by area under the curve (AUC). The sensitivity, specificity and estimated molecular testing reduction rate given the algorithm performance and dataset FGFR+ prevalence values are shown in the table.

the BLC3001 subset (420 slides). Consistent with observations from other internal studies, the performance in the curated and publicly available TCGA dataset was elevated in comparison to the real-world clinical trial datasets. In the PAN-tumor dataset, we observed a drop in AUC (0.75), auPR (0.39) and specificity (0.27), however, high overall sensitivity (0.95) remained.

The selected algorithm's sensitivity of 0.92 in the BLC3001 subset (Fig. 2), the cohort most representative of the deployment setting, was well above the targeted 0.9 sensitivity threshold, predefined as success criteria by the clinical teams. Therefore, we packaged the algorithm into a deployable Docker container, referred to as *FGFR* Device in subsequent sections, following the international standard for medical device quality management systems (ISO 13485). Once the *FGFR* Device was onboarded in the deployment platform, we moved forward with the subsequent phases of the study (Retrospective Validation, Deployment Setting Validation and Full Deployment).

Given the reported association between *FGFR*+ and certain histopathologic findings on H&E, a pathologist who is US-Board Certified in Anatomic and Clinical Pathology reviewed randomly selected true positive and true negative WSIs and tiles with highest attention score within those WSIs[33] (see Algorithm Description in the Methods section for more details on tiling of WSIs and attention models). Figure 3, A-F, shows 3 true positive and 3 true negative WSIs of MIBC from BLC3001, with a green heatmap provided by the algorithm representing normalized attention weights. True positive slides appear to harbor areas of solid tumor cell nests which is where attention weights appear to be concentrated; the morphology of the tumor on true negative slides appears more dispersed. Figure 3, G-L, shows the tiles with highest attention score from corresponding WSIs from A-F. True positive tiles show more solid tumor cells of lower to intermediate grade as compared with true negative tiles which show relatively more dispersed tumor cells of higher cytologic grade. The true positive tiles showed features in keeping with morphologies deemed typical for the few *FGFR*+ mutant tumors with histologic findings described by pathologists[33], suggesting our algorithm was selecting representative areas for the task of *FGFR*+ classification.

### Retrospective validation of the *FGFR* device for deployment
The goal of Retrospective Validation was to evaluate the analytical performance of the *FGFR* Device after onboarding in the deployment platform. The dataset was comprised of 350 (150 *FGFR*+ and 200 *FGFR*- samples; to achieve 93% power at detecting a 10% difference in sensitivity using a two-sided exact test with 5% type I error) H&E WSIs from the BLC3001 (NCT03390504) study[7] (Fig. 1). Note that these 350 samples were not used for model development and have similar characteristics as those in ANNAR, the clinical trial proposed for deployment (i.e., samples from patients with high stage disease, and images from the same five central laboratory sites).

The Retrospective Validation results are shown in Fig. 4A. The sensitivity and specificity of the proposed *FGFR* Device was assessed using the molecular test results as reference standard. We obtained an 88.7% sensitivity and 31.8% specificity. The specificity obtained was above the specified threshold in the success criteria for deployment (30%), while the sensitivity was slightly below the threshold set (90%). Also, note that the *FGFR* Device returned image quality control (QC) errors on two of the 350 slides tested. Those slides had insufficient high quality tissue tiles to perform a prediction. Tiles were deemed of enough quality based on their QC score, as explained in the Methods. Stratification by gender and age showed consistent results (supplemental Fig. 1) with those presented in Fig. 4.

Figure 4B shows a simulated clinical trial example using the obtained performance in A and assuming a 15% prevalence of *FGFR* alterations, representative of a patient population with MIBC or higher[10]. These results suggest that nearly 1 in 3 *FGFR*- patients (28.7%) could be accurately ruled out of molecular screening by using our device. Also, about 9 in 10 *FGFR*+ patients (88.7%) would still be identified for subsequent confirmatory genetic testing. Assuming a 15% *FGFR* prevalence, the implied reduction in molecular screening would be ~29% while maintaining a high sensitivity.

The central laboratory had a total of 5 laboratory sites (i.e., Geneva, Indianapolis, China, Japan, and Singapore) serving clinical sites across the globe. Figure 4C shows the ROC curves along with AUC and sensitivity values for each of the central laboratory sites. This analysis reveals that the method is not biased towards one lab, although most samples were scanned at one site (Site #1, N = 210 sample). Algorithm performance was at the 0.9 target sensitivity, within the 95% CI on sensitivity for all central laboratory sites, where the obtained point estimates were [0.9 (Site #1, N = 210), 0.9 (Site #2, N = 47), 0.87 (Site #3, N = 35), 0.77 (Site #4, N = 30), 0.9 (Site #5, N = 26)].

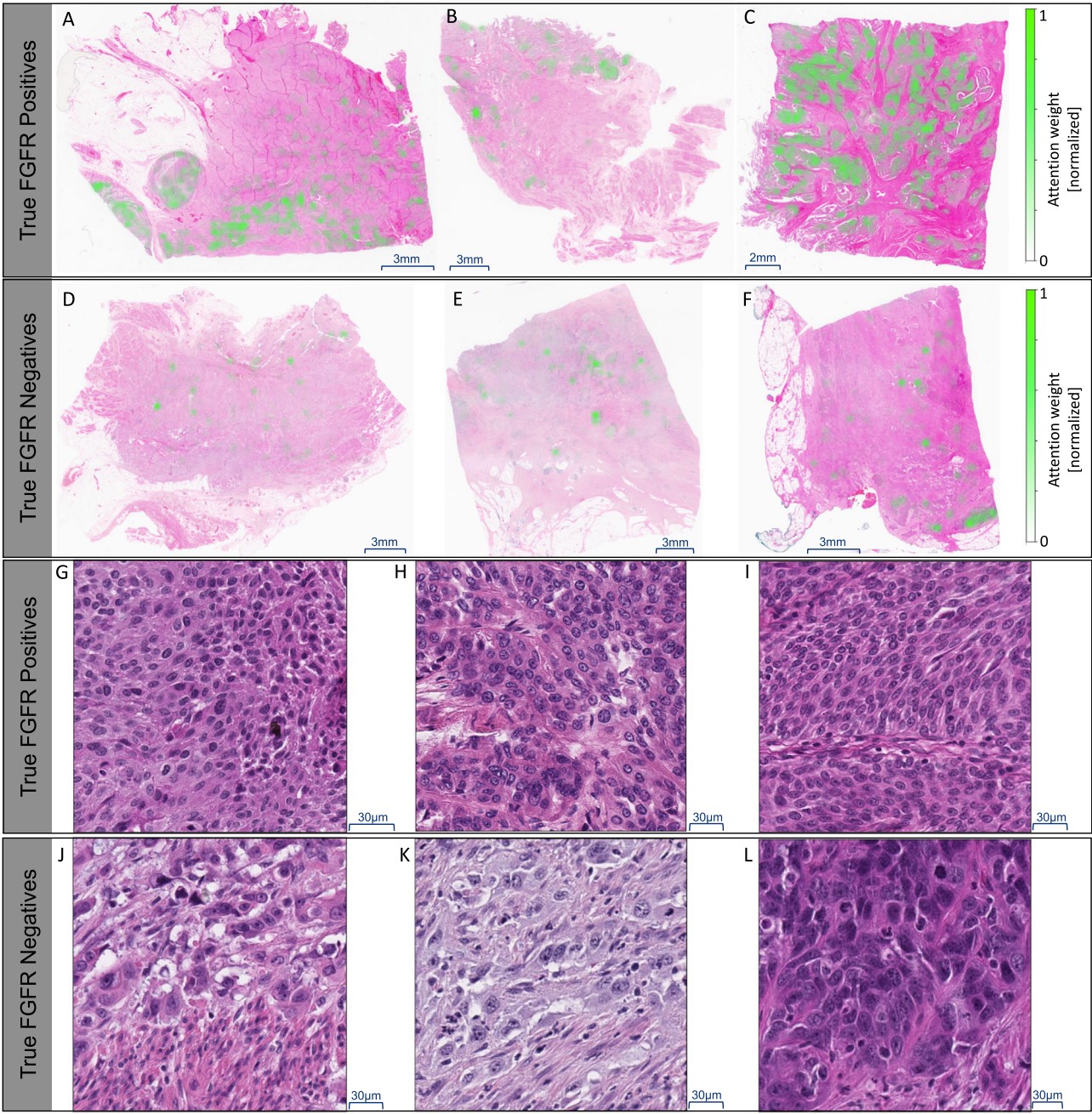

**Fig. 3 | Interpretability analysis. A–C** From BLC3001, examples of randomly selected true positive WSIs showing MIBC tumor tissue with heatmaps representing normalized attention weights (in green). The attention weights were normalized to 1. The colorbar on the right side of each panel maps to the normalized weights, all images in the panel share the same colorbar. Brighter green areas represent tiles the model deemed important to make the prediction (higher attention weight). Note that whether a slide is predicted to be *FGFR*+ or *FGFR*-, the algorithm still yields attention weights spread across all tiles used for inference on that slide. **D–F** Examples of true negative WSI with heatmaps. **G–I** Top scoring tiles at 40x magnification from the true positives shown in (**A–C**) show more solid tumor cells of lower to intermediate grade, and overall in keeping with prior observations[33]. **J–L** High scoring tiles from the true negatives shown in (**D–F**) show relatively more dispersed tumor cells of higher cytologic grade.

We then evaluated the algorithm in a simulated real-world deployment setting (outside of a clinical trial), selecting an additional "High Specificity" cut point to form a 3-tier device. The 3-tier device provided three *FGFR* likelihood levels (i.e., High, Mid, Low) based on the selected thresholds (refer to supplemental Fig. 2 for details on threshold selection). The cut point for high specificity (i.e., 0.9 specificity) was selected to create a distinction between the High vs Mid *FGFR* likelihood groups, and the cut point for high sensitivity to create a distinction between the Mid vs Low *FGFR* likelihood groups. If the predicted patient *FGFR* likelihood was High, the patient could be prioritized for molecular screening. If the predicted patient *FGFR* likelihood was Low, the patient would be deprioritized. The results are shown in Fig. 4D. Assuming a baseline population prevalence of 15%, the prevalence of *FGFR* alterations across the three tiers would be as follows: Low = 5.9%; Medium = 11.4%; and High = 49.3%, resulting in a >8 x enrichment of *FGFR*+ from the Low- to High- likelihood groups. In this scenario, nearly half of all *FGFR*+ patients in the population could be identified by testing only the 14% of patients (those considered High likelihood by the *FGFR* Device). These results demonstrate the potential clinical utility of this device for prioritizing (or de-

A) Performance on Retrospective Validation Dataset

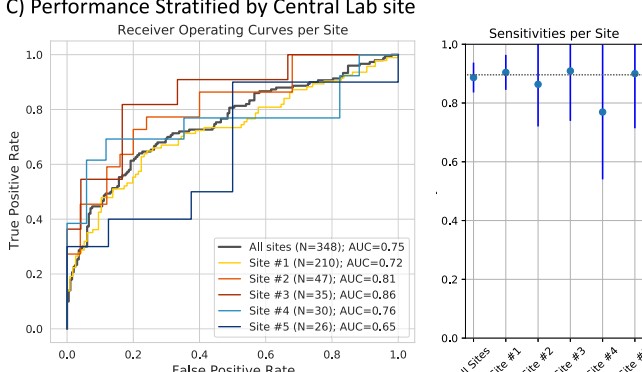

B) Simulated Clinical Trial Deployment

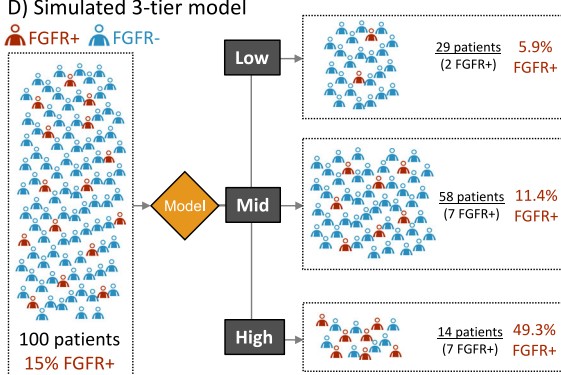

C) Performance Stratified by Central Lab site

D) Simulated 3-tier model

**Fig. 4 | Retrospective validation results. A** Confusion matrix with sensitivity and specificity metrics (target and achieved). Note this dataset was enriched for *FGFR*+ patients to achieve a statistical power of 93% (*FGFR* prevalence of 43%). **B** Simulated confusion matrix given 1000 patients assuming typical *FGFR*+ prevalence in trial of ~15%[10] and observed algorithm performance (shown in A). Note that 28.7% of patients screened by the image-based device would not be recommended for *FGFR* molecular testing. **C** Performance stratified by central laboratory site. Left-side plot shows Receiver Operating Curves (ROC) and area under the curve (AUC) values per site. The right-side plot shows sensitivity point estimates along with 95% confidence intervals (CI) per site. All sites totaled to *n* = 348 independent samples, distributed across Site #1 to #5 as *n* = 210, *n* = 47, *n* = 35, *n* = 30 and *n* = 26 respectively. **D** Simulated 3-tier *FGFR* model showing potential clinical utility for prioritizing (or de-prioritizing) patients for molecular testing in a standard clinical setting (where molecular testing may not be part of standard of care).

prioritizing) patients for molecular testing in a standard clinical setting (where molecular testing may not be part of standard of care).

## Deployment setting validation of the clinical trial workflow with the embedded *FGFR* device and full deployment for patient prescreening

The goal of the Deployment Setting Validation was to assess workflow integration of the *FGFR* Device in the proposed deployment trial. The deployment trial was the non-interventional ANNAR (NCT03955913) study[6]. ANNAR is a prescreening study to find *FGFR*+ participants with urothelial cancer (UC) for other erdafitinib studies, such as BLC3001 (NCT03390504) study[7]. The clinical workflow with the embedded *FGFR* Device is shown in Fig. 5 and is explained with further detail in the Methods section. The deployment workflow represents a "Centralized Deployment" strategy where tissue is sent to a central laboratory and tested for trial qualifying *FGFR* alterations. At the central laboratory, the tissue was also stained, scanned, and (after embedding the *FGFR* Device into the workflow) an automated tool sent the image to a cloud platform daily to run the image through the Device. The physicians then received the *FGFR* Device prediction on an online-portal and could notify the central laboratory to cancel the molecular test if desired.

The *FGFR* Device was run prospectively for 1 month on samples collected from patients diagnosed with MIBC (pT2 or higher) transferred in real time from the clinical trial, in parallel to molecular testing, using a standardized workflow. During the period of validation in real time, a total of 17 samples were received by the device. Of those 17, a prediction was generated on 15 (88%) while 2 (12%) samples could not be processed due to image QC errors. Those slides had insufficient high-quality tissue tiles to perform a prediction. Tiles were filtered based on their QC score, as explained in the Methods. The turn-around time (TAT) from receipt of images to posting results on the online-portal (to which physicians did not have access during this phase) was an average of 62 min, with a minimum TAT of 45 min and maximum of 177 min.

Due to the small number of real time samples obtained during the validation in real time (17 samples in ~1 month), supplemental retrospective samples from the ANNAR study (171 samples) were also run through the clinical workflow to calculate performance metrics. The molecular test result was used as reference, and performance is shown in Fig. 6A. Previous *FGFR* negative samples could not be used due to language contained in the ICF, so only *FGFR* positive samples were used from the time period prior to an updated ICF being approved and implemented. This resulted in an elevated *FGFR* prevalence rate in this cohort (84% *FGFR* +, compared to a reference prevalence of 15% *FGFR* in MIBC—see Discussion for additional details). The sensitivity obtained was 0.95 (with 95% confidence interval (CI) of [0.915-0.984], and the specificity was 0.25 (with CI = [0.41-0.09])), again meeting our trial team's requirements to maintain high sensitivity from an AI-based screening tool.

The algorithm was deployed for patients in 89 global study sites across 9 countries, following the international standard for device quality management systems (ISO 13485)[38]. Once live, a patient's tissue sample in WSI underwent prescreening with the *FGFR* device prior to undergoing confirmatory molecular testing. Upon receiving the results of the image-based screening, the physician decided whether to stop molecular testing. In the event that a physician did not act on the results of the *FGFR* Device, the molecular test proceeded as planned. Results during Full Deployment of the device are shown in Fig. 6B. A total of 24 samples were received by the device in the production environment. A prediction was generated for 22 (91.67%) samples and an error message (QC Failure) was generated for 2 (8.33%). The *FGFR* device predicted 5 samples as *FGFR*- (~23%) and 3 of these samples are *FGFR*- by the molecular test (1 test canceled, 1 insufficient) and thus, the model's sensitivity is 100%. The *FGFR* device predicted 17 samples as *FGFR* + (~77%) and 3 of these samples are *FGFR*+ by the molecular test (1 insufficient) and thus, the model's specificity is 19%. The molecular test was performed on 20 patients' samples and an *FGFR*+ prevalence of 15% was observed. For 3 (12.5%) samples, the *FGFR* Device provided a prediction, but the molecular test was not performed.

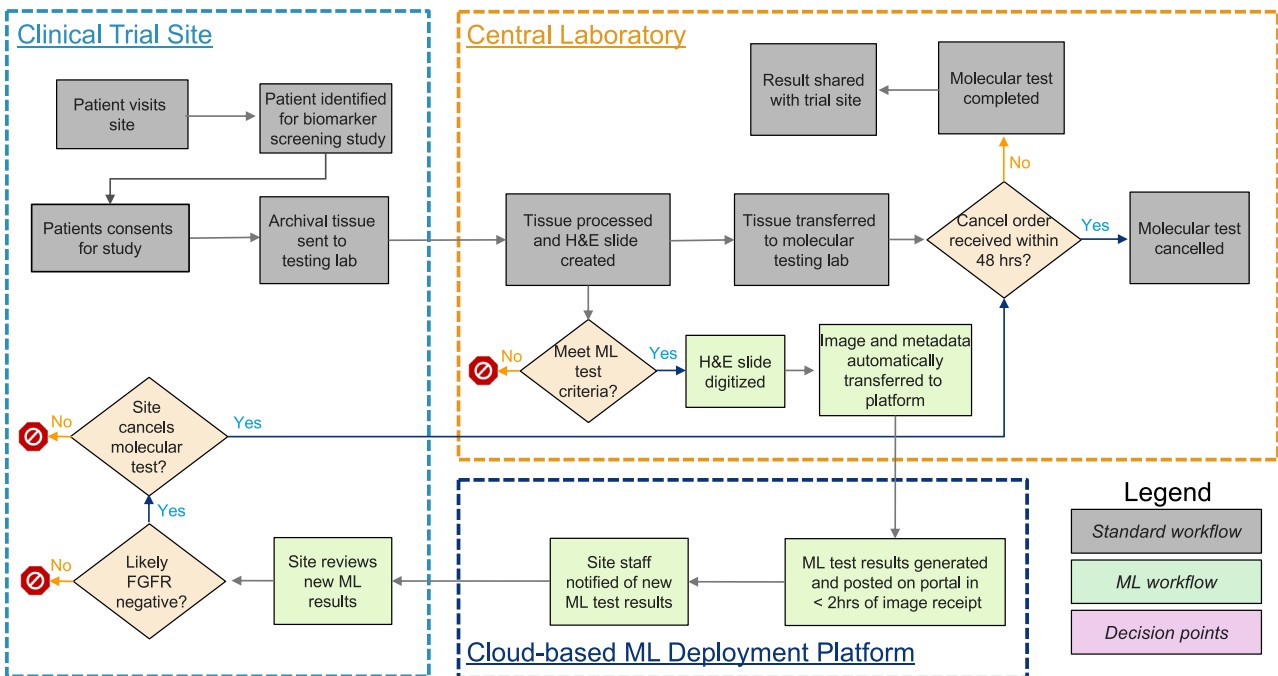

**Fig. 5 | Proposed clinical workflow for patient prescreening using the image based *FGFR* prediction device.** Clinical workflow shows three parties involved: clinical study sites, central laboratory, and cloud platform with image-based AI device. Central laboratory routinely tests tissue for clinical study qualifying *FGFR* alterations (Standard workflow). Tissue is also stained, scanned, then an automated tool sends the image to a cloud platform to run the *FGFR* Device (Machine Learning (ML) workflow). The physicians can then evaluate the *FGFR* Device prediction on an online-portal and notify the central laboratory to cancel the molecular test within 48 h.

A) Performance on Deployment Setting Validation

*FGFR Prevalence = 84%*

| | | Truth | | |
|---|---|---|---|---|
| | | − | + | Total |
| Prediction | − | 7 | 8 | 15 |
| | + | 21 | 150 | 171 |
| | Err. | 2 | 0 | 2 |
| | Total | 30 | 158 | 188 |

**Sensitivity**
Actual: 94.9%

**Specificity**
Actual: 25%

B) Full Deployment Results (~1.5 months)

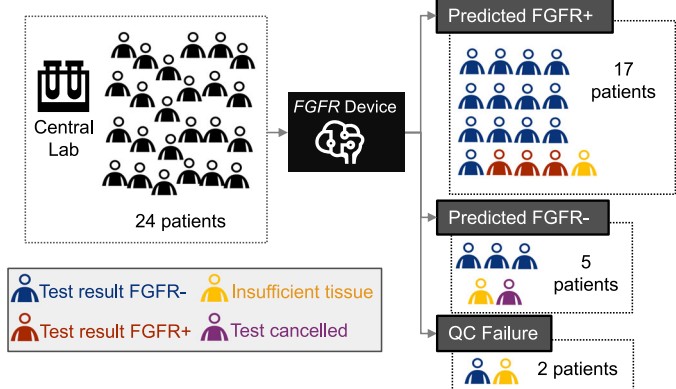

**Fig. 6 | Deployment setting validation and full deployment results. A** Confusion matrix with sensitivity and specificity metrics on real-time samples ingested from central laboratory sites (n=17) and supplemental retrospective samples from ANNAR (n=171). **B** Results during the 1.5 months of Full Deployment. The sensitivity achieved by the *FGFR* device was 100% and the specificity 19% (*FGFR* prevalence of 15% based on molecular test results). The *FGFR* device could not generate a prediction for 2 patients due to image quality errors (i.e., insufficient high-quality tiles to generate a prediction).

Notably, the molecular test was canceled by the investigator on 20% of the patients predicted *FGFR-* by the *FGFR* Device. For 9% of patients with available predictions from the device, there was not enough tissue for a confirmatory molecular test to be performed.

To demonstrate the impact of AI-based biomarker prescreening on molecular testing rates, we simulated three scenarios using performance specifications from our *FGFR* model and a range of biomarker prevalences (Fig. 7). Figure 7A shows AI prescreening with an operating point tuned to reduce molecular testing for patients likely to be biomarker negative (88.7% sensitivity and 31.8% specificity as shown in Fig. 4A), aligned with our algorithm deployed in ANNAR study sites. Molecular testing reduction rates due to AI prescreening are shown for a range of biomarker prevalences. Optimizing for AI prescreening with high sensitivity results in greater false positives but fewer false negatives; therefore, a substantial proportion of molecular tests can be avoided (~30% for low-prevalence biomarkers) while minimizing the risk that patients are incorrectly deemed ineligible for a clinical trial. We further suggest that this AI device could be used to prioritize patients for molecular testing, showing that half of all biomarker-positive patients could be detected by only testing 20% of patients when assuming biomarker prevalences≤ 15% (Fig. 7B). Finally, Fig. 7C shows potential enrichment using the 3-tier model presented in Fig. 4D. Notably, for biomarkers with low prevalence (e.g., 1%), this 3-tier approach results in an enriched ("High") cohort that is over 14

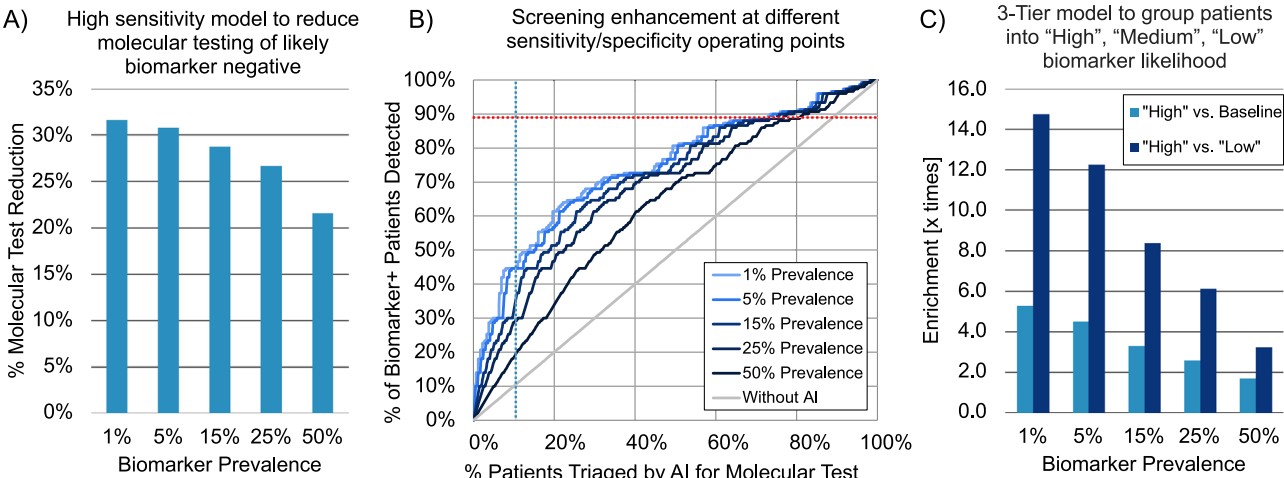

**Fig. 7 | Cost savings analysis. A** Percentage of molecular test reduction vs. biomarker prevalence using the high-sensitivity model presented in Fig. 4A (88.7% sensitivity, 31.8% specificity). In this scenario, patients predicted biomarker negative would be de-prioritized from molecular testing (i.e., rule-out patients in a clinical study with pre-defined sites like ANNAR). **B** Percentage of *FGFR*+ detected patients vs. percentage of patients receiving molecular test using different operating points of our *FGFR* model compared to without AI. Each curve represents enhancement for a given biomarker prevalence. Red and blue dotted lines represent the "Low" and "High" operating points of the simulated 3-tier model presented in Fig. 4D. **C** Prevalence enrichment using the simulated 3-tier model. Note that "High" vs. Baseline shows enrichment in the"High" group compared to baseline prevalence, whereas "High" vs "Low" shows prevalence enrichment of "High" group vs. "Low" group.

times more likely to be biomarker-positive than patients deemed to be "Low" likelihood of harboring a specific biomarker. Note the "High" likelihood cohort shows enrichment compared to Baseline prevalence as well, with up to 5 times enrichment for biomarkers with low prevalence. In this case, patients in the "High" category could be prioritized for molecular testing while testing could be deprioritized for the "Low" likelihood cohort.

## Discussion

In this study, we developed and validated a deep learning-based algorithm for inferring the presence or absence of select *FGFR* genetic alterations from H&E-stained tissues using images from large-scale bladder cancer clinical trials and real-world data sources. We demonstrated high performance on multiple retrospective datasets (i.e., AUC = 0.80 on BLC3001, BLC2002 and TCGA combined, AUC = 0.75 on PAN-Tumor data, and AUC = 0.75 on Retrospective Validation) and showed that the algorithm accurately interpreted morphologies deemed typical of *FGFR*+ mutant tumors by pathologists (Fig. 3). We also demonstrated the feasibility of deploying the algorithm in a real clinical trial to screen patients prior to molecular testing, enabling exclusion of those unlikely to harbor the targeted alterations, and demonstrate the potential impact of such AI-based biomarker models to reduce screening burden, prioritize testing resources, improve trial efficiency, and provide actionable clinical insights.

This study represents an advancement for the field in a number of ways: (1) the algorithm was developed with and validated using multiple, independent datasets, including >3000 urothelial cancer patients, allowing for a robust performance validation of an H&E-based *FGFR* prediction algorithm; (2) the algorithm was retrospectively and prospectively validated, following the international standard for device quality management systems (ISO 13485)[38], and then actively deployed in a non-interventional study[6], comprising of 89 global study sites across 9 countries; (3) the achieved performance indicates a projected reduction of 28.7% in molecular testing when deployed, which would translate to a reduction in screening burden and improved trial efficiency; and (4) the potential clinical utility for prioritizing (or de-prioritizing) patients for molecular testing in a standard clinical setting (where molecular testing may not be part of

standard of care) was demonstrated with an >8 x enrichment from Low- to High-likelihood groups.

We optimized the algorithm for high sensitivity to avoid ruling out trial-eligible patients and to mitigate any selection bias for patients recommended to continue with molecular testing. This was a requirement from the clinical and trial operations teams for the deployment of the prescreening device in an ongoing trial. Here, patients had already enrolled in the observational study and their sample was sent for molecular testing. The prescreening device was used to recommend discontinuation of molecular testing if there was a low likelihood of the patient's sample being positive for qualifying *FGFR* alterations. False-positive *FGFR* predictions would be ruled out with the molecular test itself. However, other deployment scenarios may require a threshold tuned toward specificity. If, for example, there were a limited amount of resources to dedicate to molecular testing, a group may want to use those tests on patients most likely to be identified as biomarker positive, yielding an enriched cohort for molecular testing. Figure 4D represents this scenario, with simulated results from a device that stratifies patients into three groups. Depending on resources available and varying standards of care across clinics, the physician could leverage the results of the algorithm when deciding whether to perform the molecular test or not. As you can see the probability that a patient in the "High" group is biomarker positive (~49%) is >8 x higher than patients in the "Low" group (~6%). This is the type of information we believe a physician would find invaluable when making care decisions for their patients. Additionally, the algorithm offers the added value of being able to adjust the sensitivity to fit different use cases. This flexibility allows for optimizing to various deployment scenarios and clinical decision-making processes, ultimately leading to better patient care.

The algorithm maintained a high performance on the urothelial cancer Hold-out Data; however, we observed that the model generalized well to non-urothelial tissue in the PAN-tumor dataset as well (Fig. 2), which included samples from diverse tissue sites (i.e., brain, liver, lung, prostate, skin, etc.). While we saw a slight decrease in performance (AUC = 0.75), the results suggest that *FGFR* alterations in tumor tissue might confer a shared set of morphologies across diverse tissues. Having a PAN-tumor *FGFR* model could be valuable in studies exploring *FGFR* alterations in additional tissue types and could inform

biomarker discovery efforts in those settings. Efforts to further develop biomarker models that can be applied across tumor types could benefit from inclusion of pan-tumor samples during training and experimentation with more recent analytical techniques.

While our algorithm was initially designed as a multi-instance learning attention-based network with a CNN backbone[39], the rapid evolution of technological advancements in computer vision methodologies has outpaced the time required for proper development, validation, deployment in a global clinical study, and drafting this manuscript to share our experience with the scientific community. Notably, newer methods such as vision transformer networks have emerged as alternatives to CNNs[40], with the potential to offer increased performance, especially when trained on smaller datasets[41]. Additionally, self-supervised learning (SSL) has also shown promising results in the field of histopathology, enabling models to be more generalizable across scanners, staining procedures, and tissue types[42,43]. In our recent work[43], we demonstrate how pre-training the CNN of the model mentioned in this manuscript using a large unlabeled dataset (consisting of 25 k WSIs from multiple scanners, hospital systems, disease stages, and tissue sites) via SSL resulted in a more generalizable model with improved performance in detecting FGFR+ in non-muscle invasive bladder cancer (NMIBC) and pan-tumor WSIs.

Despite these advancements in performance, the interpretability of AI-based biomarker tools still remains a significant challenge. Though some work has been done in this space[33], general pathologists have not been trained to infer the presence of an FGFR alteration (or others) from an H&E-stained slide, and thus should not be considered as a point of reference on which to evaluate algorithm performance. Nevertheless, insights from a pathology standpoint can be gleaned from the algorithm's attention scores, which can highlight regions of the WSI deemed to be more (or less) informative in the algorithm's biomarker prediction (Fig. 3). Here, positively predicted WSIs tended to have tiles showing solid tumor nests of lower cytomorphologic grade, while negatively predicted WSIs tended to have tiles showing dispersed tumor nests of higher cytomorphologic grade, overall in keeping with limited prior descriptions by subspecialized pathologists[33]. Because, in contract to tumor-detection algorithms, results from AI-based biomarker detection tools cannot be confirmed by a human pathologist viewing the WSI, these tools must be robustly designed and tested in a thoughtful manner to ensure that physicians can confidently incorporate these insights into their care decisions.

Here, we present a staged validation strategy leveraging multiple independent datasets and incorporating decision-points throughout the process. In collaboration with clinical and regulatory stakeholders, we developed a Retrospective Validation study with two key factors in mind: (1) the study be sufficiently powered to reliably confirm the performance of the device relative to pre-specified success criteria, (2) the study be representative of the planned deployment setting in the ANNAR study. The BLC3001 study offered a dataset that enabled a well-powered and representative validation study, so a sufficient number of samples (150 FGFR +, 200 FGFR-) were set aside after power calculations for our Retrospective Validation step, following guidelines from the international standard for device quality management systems (ISO 13485)[38]. To ensure the integrity of these samples, the Retrospective Validation step was only performed after a model had been selected, locked, and packaged for deployment. During model development, we set aside parts of our development data sets (i.e., Hold-Out Data) for evaluating performance after training a model with cross-validation. Through discussion with various stakeholders, a final model was selected amongst several distinct, high-performing models based on performance in this Hold-Out Data, at which point it was locked and we proceeded to Retrospective Validation. After the Retrospective Validation step, we pursued further validation in the planned Deployment Setting using data from the ANNAR study. This included two parts: (1) further evaluation of performance based on patients enrolled

in the ANNAR study, and (2) live deployment into the sample workflow to identify and overcome any technical hurdles that may arise. While retrospective samples collected from the ongoing ANNAR study would have been ideal for additional validation, the ICF only allowed for analysis of images collected from FGFR+ patients. In order to deploy the AI-based tool on patients who tested FGFR- or who had yet to receive a molecular test, an updated ICF was submitted to and approved by local review boards. As a result, our Deployment Setting validation step included significantly more FGFR+ samples than would be expected in a standard population, but nonetheless enabled a confirmation of the Sensitivity of the algorithm, a high priority for project stakeholders. This step also provided an opportunity to test and optimize the real-time data flow as the device was embedded into the existing workflow.

The deployment workflow in this study (Fig. 5) was adapted to seamlessly embed into the existing lab workflow, and optimized to return results fast enough to enable decision making and readjustment on the fly (average TAT of ~1 h). The primary bottleneck was on the image upload from the central laboratory to the deployment cloud, which was only performed once per day, at the end of each business day. Nevertheless, the turn-around-time for image-based device results in this setting was <24 h, and the investigators were allowed a time window of 48 h from image generation to stop the molecular test workflow. If the investigator did not reply to the query in that period, the workflow for molecular testing continued to avoid delays in molecular testing or study enrollment. Incorporating the FGFR Device into the existing workflow also offered another benefit; it delivered insight into the FGFR status for patients where molecular tests could not be carried out due to a lack of sufficient RNA or tumor tissue. This strategy could prove effective for saving resources associated with molecular testing, by ruling out patients likely to be biomarker negative; however, other deployment strategies could provide other benefits.

While we chose to deploy the algorithm at a central laboratory to optimize for the number of patients screened with the algorithm, future efforts are likely to employ a 'decentralized' strategy in which the algorithm is deployed locally at each individual clinical site. In that scenario, tissues slides and images would be scanned and generated locally, avoiding the need to ship tissue to the central laboratory. While the complexity of deployment increases, the clear advantage of the decentralized strategy is a reduction in time to insight from the algorithm. Even though we optimized for rapid return of algorithm results to physicians in central deployment, we could not get around the time spent shipping the sample to the central laboratory. In a decentralized deployment setting, the algorithm could be leveraged locally by physicians to inform whether to enroll a patient or ship the tissue at all. Furthermore, it would allow having a more streamlined workflow, given the heterogeneity of site-level paradigms for reporting back algorithm results. One challenge in our deployment process was the requirement to use two different web portals, one for reporting algorithm results and the other for evaluating molecular test results and reporting test cancellations, which was already in use by clinical sites at the central laboratory. This may have added an extra layer of difficulty for clinical site investigators at the time to report test cancellations based on the algorithm's predictions. One could easily imagine a scenario where this type of biomarker-detection algorithm is being used ubiquitously across all qualifying patients at a health system at which the device is deployed, providing physicians with fast, actionable insights that can inform treatment or testing decisions in real time.

Despite the importance of biomarker testing in clinical care and trial enrollment for bladder cancer patients, uptake of molecular testing can be limited by high testing cost, limited assay availability, and slow turn-around times (especially outside academic medical centers). These challenges are compounded when biomarker prevalence is low and high testing coverage is needed to ensure

biomarker-positive patients are properly matched to optimal therapies. Using the AI performance (i.e., ROC curve) obtained in our Retrospective Validation (Fig. 4C), we sought to better understand the potential impact of AI-based biomarker screening tools (Fig. 7). Our analysis showed that with AI prescreening, a >30% reduction in molecular testing during trial screening can be achieved for rare biomarkers while minimizing the risk of patients being incorrectly categorized as biomarker negative and ineligible for the trial (Fig. 7A). This AI-enabled reduction in molecular testing can translate to substantial cost savings for healthcare systems treating patients or pharmaceutical companies running clinical trials. For example, in 2019 an estimated 80,500 patients were diagnosed with bladder cancer United States, 30% of whom (~24,000) presented with muscle invasive cancer[44]. Approximately 15% of those patients (3,600) are likely to be *FGFR* +, but performing NGS testing on all patients (assuming a cost of \$5000 per test[45]) would result in ~\$120 million for molecular testing (~\$33,000 per *FGFR*+ patient identified). However, our results show that an AI-based screening device could result in ~\$35 million in savings annually while maintaining high (~90%) sensitivity, resulting in a molecular testing cost of ~\$26,000 per *FGFR*+ patient detected. Additionally, our analysis shows that AI prescreening for molecular testing can yield significantly enriched patient cohorts that could enable detection of more biomarker-positive patients with fewer tests (Fig. 7C). We show that prioritizing molecular testing based on an AI-screening tool could enable detection of approximately half of all biomarker-positive patients by only testing ~15–20% of patients (Fig. 7B). This suggests that the first ~1800 *FGFR*+ patients could be detected for ~\$20–22 million (~\$12,000 per *FGFR*+ patient), a significant improvement over the ~\$60 million it would require to identify 1800 patients if testing was performed at random (as is currently done with no enrichment tool available). Notably, the remaining ~1800 *FGFR*+ patients would require nearly \$100 million in molecular testing to uncover (~\$55,000 per *FGFR*+ patient). AI biomarker prescreening could serve as a cost-effective method for expanding testing to patients who are likely to be biomarker positive but who would otherwise not have received biomarker testing due to lack of access or limited tissue availability.

This work constitutes a milestone in the implementation of AI-based screening in the clinical setting. We demonstrate a robust validation and deployment of AI-based screening tools in a clinical setting. The deployment of this algorithm could increase access to care for patients with *FGFR*-driven diseases, especially in areas where *FGFR* inhibitors are approved for use. Additionally, we demonstrate the potential economic impact of AI-based biomarker detection algorithms for enriching patient populations in a clinical and drug development setting. Most importantly, with this type of algorithm, we believe physicians could gain rapid, actionable insights into a patient's specific disease and make more informed care decisions in a timely and efficient manner, resulting in improved patient outcomes and a higher quality of life.

## Methods

### Ethics approval and consent to participate
The study was approved by ethics review boards from 89 sites participating in the ANNAR study, NCT03955913. These sites were sponsored by Janssen R&D, the entity that provided global oversight and approval of the study. The study was carried out in accordance with relevant legislation and ethics guidelines. Enrolled patients provided informed signed consent prior to participating in the study.

### Study design and datasets
We collected data from public repositories, third-party vendors, and internal clinical trials amounting to 3940 histology images (H&E-stained whole-slide images of urothelial carcinomas). These images were linked to ground truth molecular testing results, either by NGS or targeted assay. Image ground truth (*FGFR* positive or *FGFR* negative) was defined by determining whether the *FGFR* alterations from the QIAGEN Therascreen® *FGFR* RGQ RT-PCR Kit, which aids in identifying patients eligible for treatment with BALVERSA™ (erdafitinib). The alterations are the following: (1) *FGFR3* gene point mutations with targets p.R248C (c.742 C > T), p.G370C (c.1108 G > T), p.S249C (c.746 C > G), p.Y373C (c.118 A > G); (2) *FGFR3* fusions with targets TACC3v3, TACC3v1 and (3) *FGFR2* fusions with target BICC1 and CASP7.

To develop the algorithm, we used one whole slide image from bladder tissue per patient from three different cohorts: 407 from The Cancer Genome Atlas (TCGA) consortium (https://portal.gdc.cancer.gov/projects/TCGA-BLCA), 2811 from BLC3001 (NCT03390504) and 184 from BLC2002 (NCT03473743), two erdafitinib trials[7,8], as seen in Fig. 1. The prevalence of *FGFR* in each cohort was 12.5%, 11.6% and 15.7% respectively, totaling a ~12% average prevalence. The Development Data was split into Training Data (85%, or 2820 slides) and Hold-out Data (15%, or 582 slides). The data split preserved the same ratio of *FGFR*+ vs. *FGFR*- patients, as well as the proportion of samples from each cohort. The Training Data set was used for algorithm optimization via cross-validation, and the Hold-out to evaluate performance prior to algorithm packaging for onboarding on deployment platform, Retrospective Validation, Deployment Setting Validation, and Full Deployment. The Train Data was further divided into 5-folds to perform cross-validation for hyperparameter tunning. Similarly, the *FGFR* mutation and cohort ratios were preserved in each fold.

Note that an extra subset of samples from the BLC3001 (NCT03390504) cohort (350: 150 *FGFR* +, 200 *FGFR*-) were left out for Retrospective Validation of the *FGFR* Device for deployment. To achieve the desired confidence intervals calculated via statistical power analysis for the estimated sensitivity and specificity (to detect a 10% difference in sensitivity using a two-sided exact test with 5% type I error), a total of 150 samples was randomly selected from the positive dataset population, and 200 from the negative dataset population. Furthermore, data from ANNAR (NCT03955913)[6], the deployment trial, was used for the Deployment Setting Validation of the *FGFR* Device (17 WSIs acquired in real time for workflow validation and 171 retrospective WSIs to assess performance). An additional independent test dataset (361 WSIs) from an external laboratory with tissue from multiple tumors (i.e., PAN-Tumor) was used to evaluate generalization of the algorithm to solid tumors (not represented in the figure).

### Algorithm description
Deep learning methods were used to predict *FGFR*+ based on an H&E-stained histopathology slide. Specifically, we used convolutional neural networks (CNNs), which excel at pattern recognition for data with inherent structure, like images or sequences. For more efficient training, we incorporated transfer learning[46] into our approach. That is, we used CNNs that had been previously trained on more general image data sets to recognize simple and complex patterns, which allows them to be more quickly tuned to new, related tasks such as classifying histopathology images.

Additionally, we developed a multi-instance learning approach to accommodate the exceptionally large images obtained by scanning histopathology slides[39,46,47]. In this framework, a whole histopathology slide is broken into many smaller tiles. The patient-level outcome associated with the slide is associated with each individual tile during training and the network learns patterns that differentiate the patient label (e.g., *FGFR*+ or *FGFR*-). This approach has the added benefit of not requiring manual annotation of the whole-slide image by a pathologist, resulting in a lower cost of obtaining data and a broader set of outcomes on which to train. Figure 8 shows the multi-instance learning pipeline embedded in the *FGFR* device. Note that all tiles in the slide are fed into the network to predict a single outcome (i.e., *FGFR*+ or *FGFR*-) for the entire slide.

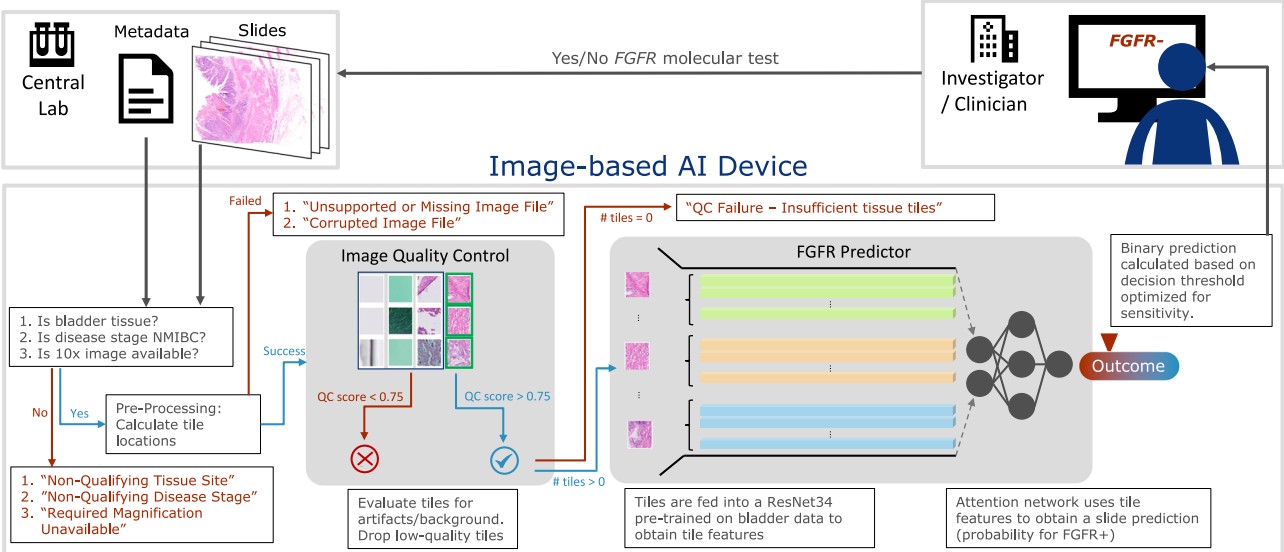

**Fig. 8 | *FGFR* device overview.** The device is a Docker container with the algorithm combined with error checks that allowed for easy integration into the clinical workflow. It takes an image along with the corresponding metadata as inputs, and outputs the likelihood of *FGFR* for that image. The device will show explanatory error message for the clinician when slides don't meet pre-specified criteria (i.e., tissue site must be bladder, from MIBC disease stage, and 10 x magnification available). Similarly, it will notify the user if the image does not pass quality control (i.e., image is corrupted or missing, or there are insufficient high-quality tiles to perform a prediction). These checks ensured that the device would only run-on data of same distribution as the one used for training.

Whole slide images were preprocessed into 224 × 224 pixel non-overlapping tiles to train the multi-instance learning pipeline. The tiles were fed into the quality-control pipeline[48] to remove the tiles with artifacts (i.e., pen marks, blur, etc...), followed by a stain-based data augmentation step[49] to generate multiple stain versions of each tile for training. Tiles where the quality control (QC) score was below 0.75 were dropped (see device description in Fig. 8 and provided pseudo-code from Box. 1 for detailed QC steps). Given that similar performance was obtained at multiple magnifications available, we decided to train on images at 10 x magnification to speed up algorithm training and inference time. A CNN followed by an attention-based network[39] was trained end to end using the multiple-instance learning pipeline[50]. The CNN initial weights used were from an ImageNet pre-trained ResNet34 network from the pyTorch library in Python. The attention network started with random initialized weights. We performed hyper parameter tuning via grid search and used a cross-entropy weighted loss function to offset the class imbalance between positive and negative *FGFR* slide count. We selected the best algorithm as being the one with the highest positive predictive value (PPV) at 0.9 sensitivity on the validation sets from a 5-fold cross-validation data split. The optimal hyperparameters were found to be the following: learning rate of 0.00001, weight decay of 0.0001 and dropout of 0.5.

**Proposed clinical workflow**

Figure 5 shows the proposed workflow for patient prescreening using the image-based *FGFR* Device. The device is used prior to planned molecular testing to identify subjects in whom molecular *FGFR* testing is likely to be negative. Results of digital device analysis are provided to inform clinical trial investigators and help them screen patients who could be eligible for the clinical trial and prioritize patients for molecular testing. Note there are three parties involved in the workflow. The clinical study sites, which are distributed around the globe, the central laboratory, which has multiple central locations (i.e., Indianapolis, Geneva, Japan, and Singapore) in contact with its corresponding investigator sites, and the cloud platform partner, which is connected to the central locations from the lab and the investigators sites.

The gray boxes represent the workflow steps that were being followed for patient enrollment prior to the implementation of the image-based AI prescreening. Starting at a clinical trial site, a patient that meets the enrollment criteria for the trial would sign consent to enroll, and then archival tissue of a tumor biopsy would be sent to the central laboratory for H&E staining and scanning. After quality control of the tissue, the CRO would then send it to genomics to perform a molecular test (i.e., QIAGEN therascreen® *FGFR* RGQ RT-PCR Kit) to identify if the patient is *FGFR* +, and hence, eligible for treatment with BALVERSA™ (erdafitinib).

The green boxes represent the steps added to the prior workflow to add the image-based prescreening. Starting at the central laboratory, the staining and imaging department performs a daily transfer of the scanned images and corresponding metadata (i.e., patient id, slide id, tissue site of specimen, etc.) to the cloud platform hosting the *FGFR* Device. The device runs as soon as the images reach the platform, and in a matter of minutes, the results of the algorithm (i.e., *FGFR* likelihood) are available via web portal to the investigators. Investigators receive an email notifying them an *FGFR* result is available for them to review, and based on the result, they decide whether to cancel the molecular test. In that case, they notify the central laboratory by answering a query in their portal.

**Design control development and validation of *FGFR* device**

As determined by our regulatory and clinical diagnostics teams, the algorithm was classified as Software as Medical Device[38]. As a result, we applied medical device standards, including design controls, to the development and validation process. The development, design verification and design validation steps that were followed explained in more detail below. The decision to move forward to fully deploy the algorithm was based on the results of a Retrospective Validation study using representative samples (section C), and a Deployment Site Validation study in which the device was deployed on prospectively collected ANNAR samples (section D).

**A. Algorithm packaging and software verification**

The first step after algorithm training and optimization using the 3402 slides was to package the selected algorithm into a user-friendly device (see schematic in Fig. 8). The device is a Docker container with the algorithm combined with error checks that allowed for easy

**BOX 1**

# Pseudocode representation of the FGFR device (shown in Fig. 8)

1. Parse input arguments (*input_WSI_path, tissue_site, disease_stage*).
2. Input metadata QC:
 2.1 if *tissue_site* is not "Bladder": return *"Non-qualifying tissue site."*
 2.2 if *disease_stage* is not "MIBC": return *"Non-qualifying disease stage."*
3. Read WSI:
 3.1 if *input_WSI_path* is empty: return *"Unsupported or missing image file."*
 3.2 if OpenSlide returns read error: return *"Corrupted image file."*
 3.3 if 10x magnification is not available: return *"Required magnification unavailable."*
4. Run image quality control (QC) filters:
 4.1 Obtain low-resolution thumbnail to allow faster preprocessing.
 4.2 Compute pen marks and background binary masks:
 4.2.1 Red, green, blue pen marks:
 *filter.filter_green_pen(), filter.filter_blue_pen(), filter.filter_red_pen() ***
 4.2.2 Background:
 *filter.filter_grays()**
5. Calculate tile (x, y) locations given tile dimension [224×224] and 10x magnification.
6. For each tile, calculate the image QC score [0,1]:
 6.1 Compute percentage of tissue in tile using output masks from 4.2:
 *tissue_percent, quantity_factor = tiles.tissue_quantity_factor(tissue_quantity()) ***
 6.2 Compute color_factor and saturation_factor measurements in tile:
 *color_factor = tiles.hsv_purple_pink_factor(), saturation_factor=tiles.hsv_saturation_and_value_factor() ***
 6.3 Calculate QC score:
 *score = 1–10/(10+tissue_percent$^2$·ln(1+color_factor·saturation_factor·quantity_factor)/100) ***
7. Keep tiles with *score* > 0.75:
 7.1 if number of remaining tiles (*N*) is <1: return *"QC Failure – Insufficient tissue tiles."*
8. Run *N* tiles through a *ResNet34* convolutional neural network** to extract feature vectors of size *N*x512.
9. Run feature vectors through an attention network[39] to obtain the WSI likelihood of *FGFR*.
10. Threshold the outputted likelihood to binarize result as *FGFR+* or *FGFR-*
*\* Functions are from open-source code found in utils.py, filter.py and tiles.py from[48]*
*\*\* Network structure available in* https://pytorch.org/vision/main/models/generated/torchvision.models.resnet34.html

---

integration into the clinical workflow. The device takes an image as input along with the corresponding metadata for that image, and outputs the likelihood of *FGFR* for that image. In cases where the slide metadata does not meet the predefined criteria (i.e., bladder tissue, 10 x magnification image, MIBC) the device will show an explanatory error message for the clinician. Similarly, it will notify the user if the image does not pass quality control (i.e., image is corrupted or missing, or there are insufficient high-quality tiles to perform a prediction). These checks ensured that the device would only run-on data of same distribution as the one used for training.

### B. Migration of device to deployment platform

After development, packaging for deployment (i.e., Docker) and software testing under design controls, the device was shared with our deployment partner to embed it in their cloud platform. First, the deployment partner run the device on their cloud platform to evaluate the fidelity of the device and ensure that the performance metrics agreed with those obtained during software verification. Then, the cloud platform was integrated with the clinical workflow by connecting the central laboratory sites to Amazon Web Services S3 data ingestion buckets, and by means of a web portal for investigator sites around the world participating in the trial to show the *FGFR* device predictions.

### C. Retrospective validation

As mentioned in the Data section, a retrospective data set comprised of 350 (150 *FGFR+* and 200 *FGFR-* samples; to achieve 93% power at detecting a 10% difference in sensitivity using a two-sided exact test

with 5% type I error) representative H&E histopathology images were designated for the planned retrospective design validation phase. These samples were not utilized to train the algorithm and were not accessible during development (tuning/training/initial testing) of the algorithm. The sensitivity and specificity of the *FGFR* Device was assessed using the QIAGEN molecular test as the reference standard.

The acceptance criteria to determine if the *FGFR* device would be deemed suitable for prospective design validation was by stakeholders as follows: if the point estimate (PE) of sensitivity ≥90%, lower bound (LB) 95% CI ≥ 80%; and PE of specificity ≥30% with LB 95% CI ≥ 20%, the device would move forward to prospective validation. Otherwise, it would be an active decision, and including analytical performance in the acceptance criteria of the prospective validation (section D) could be considered.

### D. Deployment setting validation and full deployment

The goal of the Deployment Setting Validation was to assess and optimize workflow integration of the *FGFR* Device in the ANNAR study. The device was deployed in the ANNAR study in parallel to the QIAGEN molecular test. The results of the algorithm were not reported to investigators at this stage since the objective was to demonstrate clinical study workflow integration and concordance with the molecular test.

The device was run on ~1 month worth of prospectively collected ANNAR samples transferred in real time (17 samples), in parallel to molecular rest, using a standardized workflow to mimic full deployment workflow. The metrics captured were the % of images successfully completing workflow and the turn-around-time (TAT) from

receipt of images to posting results on physician portal. The device was also run on supplemental retrospective samples to measure sensitivity and specificity, as an exploratory analysis.

The acceptance criteria to proceed to full deployment was a TAT of all samples on which the device was successfully run (a prediction or error is generated) of <24 h and a holistic review of performance data on retrospective and prospectively collected datasets by internal stakeholders. The validation study was conducted under an Investigational Device Exemption (IDE) regulatory designation.

Under the intended use of the device for full deployment, a patient first underwent screening with our image-based device prior to undergoing molecular testing. Upon receiving the results of the image-based screening, the physician had the choice to stop the molecular testing. Enrollment into subsequent, interventional clinical trials was contingent on the confirmed *FGFR*+ status, based on molecular testing.

### Reporting summary
Further information on research design is available in the Nature Portfolio Reporting Summary linked to this article.

## Data availability
The raw data used in this article was collected from multiple Janssen R&D (Johnson & Johnson) clinical studies (NCT03955913, NCT03390504, NCT03473743, NCT03955913) where data was approved for research use. This raw data and the study protocols are not publicly available due to reasons of data sensitivity, including research participant's privacy/consent. Inquiries about clinical study raw data may be made to the authors (ajuanram@its.jnj.com), although access is subject to permission of the corresponding data owners for each of the Janssen R&D (Johnson & Johnson) clinical studies listed above. We also used public raw data from The Cancer Genome Atlas (TCGA) consortium for development. This data is publicly available at https://portal.gdc.cancer.gov/projects/TCGA-BLCA. The processed source data and code required to reproduce the results presented in this manuscript are publicly available at https://github.com/johnsonandjohnson/FGFR_Device_Review.

## Code availability
We provide a technical description of the *FGFR* device in the online Methods, together with Fig. 8 depicting the device structure and pseudocode in Box. 1 to facilitate the understanding of the deep learning algorithm for biomarker prediction from H&E WSIs. This includes a description of what sections of the pipeline were based on open-source code, available at https://github.com/CODAIT/deep-histopath/tree/master/deephistopath/wsi. The full code base from the FGFR device is not publicly disclosed to safeguard Janssen R&D intellectual property. Access requests for such code will not be considered to safeguard Johnson & Johnson Innovative Medicine's intellectual property. However, access to predictions and source code for data analyses and figure generation in this work are publicly available and can be downloaded from https://github.com/johnsonandjohnson/FGFR_Device_Review.

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

## Acknowledgements

The authors wish to express their gratitude to Songbai Wang, Harinder Chhabra, Christopher Medberry, Cheng Zhang, Adam Yala, Najat Khan, the Johnson & Johnson R&D Computer Vision, ANNAR Clinical Operations, Oncology Diagnostics, Oncology Data Science, and Data Science Platform teams for their contributions and support throughout this project and manuscript preparation. Their expertize and dedication greatly contributed to the project's success. We appreciate their involvement and positive impact.

## Author contributions

A.J.R. and C.P. were responsible for the development of the algorithm and the quantitative analyses of the data. C.C., C.P. and A.J.R. were responsible for the software verification and packaging. P.R. provided histopathology feedback. O.C.Z., A.B. and A.J.R. lead the algorithm validation and deployment in trial. N.S., S.T., M.Q., P.C., K.S., S.Y., O.C.Z., T.M., J.G. and A.J.R. were responsible for the study design and planning. K.S., T.M. and J.G. were stakeholders overseeing the project. All authors jointly planned, wrote, reviewed, and approved the manuscript.

## Competing interests

All authors are currently, or were formerly, full time employees of Janssen Research & Development (Johnson & Johnson). They are, or were formerly, salaried and owners of stock in Johnson & Johnson companies. PR owns Paige stock and is a member of the College of American Pathologists' Digital and Computational Pathology Committee.
