## [Peer Review File · Nature Communications]

REVIEWER COMMENTS

Reviewer #1 (Remarks to the Author): Expert in bladder cancer targeted therapy, genomics, biomarkers, and FGFR

Ramon et al perform an analysis of H&E digital images with paired genomic data taken from a range of sources including public repositories and commercial clinical trials including FGFR inhibitors. They outline the discovery of a FGFR related morphological biomarker in a large cohort (>3000 patients) using a deep learning techniques and validate this in a separate retrospective patient cohort enriched for FGFR+ patients. The validated biomarker was then prospectively deployed in a clinical trial setting with exploratory outcomes including total analysis time of the device and influence on clinical decision to proceed to molecular testing.

This is an extremely well written manuscript which follows the biomarker road map (and medical device standards) and is logically presented. The statistical analyses seem sound and appropriate conclusions have been reached. The particular strengths are the large sample numbers as part of the discover data set and the attempt to prospectively assess in a non-interventional clinical trial. Patient selection of FGFR inhibitors for patients with urothelial cancer (and other tumours) is of key importance and this manuscript will be of interest to a wide audience including oncologist, pathologists, computer scientists and biomarker developers. This is the most thorough attempt at deploying a FGFR digital pathology algorithm and as such would strongly recommend publication in nature communications. However given that poor specificity, the main utility of the device in its current form is quickly identifying patients that are unlikely to be FGFR+ and therefore start an alternative therapy without delay. I think this will have a minimal impact on the number of genomic tests performed and overall patient management. Greater insights into how the test could be further improved would be useful to the community. There are some minor comments that may improve the manuscript for readers.

1) The presentation of the ANNAR dataset in Figure 1 is misleading. It depicts this study as prospective validation whereas 171 patients in the trial were used as part of the retrospective validation. Why were the data from BLC3001 and ANNAR kept separate rather than combining as a single retrospective validation set?

2) Throughout the manuscript, there is mention about prospective validation of the device – eg paragraph 5 of the introduction. Given that this was insufficiently powered to show a statistically significant difference, it is unable to validate the findings. The term used in Figure 1 is more accurate as ‘a deployment trial’ and this should be made clearer earlier in the text of the manuscript.

3) The device positivity threshold has been set with a >90% sensitivity but this is at the cost of specificity which is poor around 25%. There is no mention in the discussion about what threshold is deemed clinically relevant. Notably, the simulated 3-tier model in figure 4 compares low and high device

predictions and avoids the mid-risk prediction which predicts an 11.4% positivity, a similar rate of positivity to BRCA+ in prostate cancer which is now routinely performed in many countries.

4) In the discussion, there is no mention about how independent the H&E FGFR device output is compared to conventional and histopathological characteristics. For example in UC, platinum sensitivity is associated with FGFR+ and in pan-tumour analysis sex, race, and histological variant has been associated with FGFR status. Were these clinical variable reviewed and could they be integrated into a model to improve the positive predictive value?

5) With a set of rules, can a trained pathologist denote FGFR+ with similar accuracy?

Reviewer #2 (Remarks to the Author): Expert in deep learning for digital pathology, whole-slide imaging, and cancer therapy

In their manuscript „Development and deployment of a histopathology-based deep learning algorithm for patient pre-screening in a clinical trial for targeted-therapy in bladder cancer” Ramon et al. trained a deep learning model on image data of histopathological slide to predict FGFR-Alterations in bladder cancer. They used different training and validation cohorts originating from the TCGA-BLCA project and different pharmaceutical trials.

While the overall topic is interesting, there are several major drawbacks of the manuscript in my opinion:

1. Novelty

There are several studies which have used almost the exact same research question, similar methodology and achieved comparable results. For example ref. 33 also used the TCGA dataset to predict certain FGFR alterations from H&E slides using conv nets. The performance of the current study is comparable ref. 33 despite using substantially more slides. What is truly new and unique about the authors approach? Including two new types of alterations in the same signaling axis does not really count as novelty to me. Neither does using more slides, without substantially improving performance. Similar studies include work done by Velmahos et al., Woerl et al., and others.

Velmahos CS, Badgeley M, Lo YC. Using deep learning to identify bladder cancers with FGFR-activating mutations from histology images. *Cancer Med.* 2021 Jul;10(14):4805-4813. doi: 10.1002/cam4.4044. Epub 2021 Jun 10. PMID: 34114376; PMCID: PMC8290253.

Woerl AC, Eckstein M, Geiger J, Wagner DC, Daher T, Stenzel P, Fernandez A, Hartmann A, Wand M, Roth W, Foersch S. Deep Learning Predicts Molecular Subtype of Muscle-invasive Bladder Cancer from Conventional Histopathological Slides. *Eur Urol.* 2020 Aug;78(2):256-264. doi: 10.1016/j.eururo.2020.04.023. Epub 2020 Apr 27. PMID: 32354610.

Zhang X, Wang J, Lu J, Su L, Wang C, Huang Y, Zhang X, Zhu X. Robust Prognostic Subtyping of Muscle-Invasive Bladder Cancer Revealed by Deep Learning-Based Multi-Omics Data Integration. *Front Oncol.* 2021 Aug 6;11:689626. doi: 10.3389/fonc.2021.689626. PMID: 34422643; PMCID: PMC8378227.

2. Methodology / Data

The algorithm which was used (Pretrained Conv Net with MIL) is somewhat outdated and there are newer methods (i. e. vision transformer) available. Furthermore, it is difficult to really assess the underlying code base as it has not been made publicly available. This is unusual. What is the reason for not providing the code and models a) for review purposes and b) as open source to the scientific community? Particularly as the model has been trained on data funded by the public and assumingly with open source methods?

Although the authors should be commended for using various test and validation data sets, I'm confused about the exact terminology. As far as I'm aware, the term "testing data" should be reserved for unseen data from an external source to make the final performance estimation. Fig. 1 somewhat indicates a different use. Furthermore, it seems to be the case that some patients from the "Development Datasets" (specifically 350 WSI from the BLC3001 trial) have been used as "Retrospective Validation". This is not standard. Why didn't the authors use the BLC2002 trial or the TCGA cases as "Retrospective Validation"? Assuming characteristics about future cases the model should be used on and tweaking the data accordingly is problematic (see Kleppe et al.).

Kleppe A, Skrede OJ, De Raedt S, Liestøl K, Kerr DJ, Danielsen HE. Designing deep learning studies in cancer diagnostics. *Nat Rev Cancer.* 2021 Mar;21(3):199-211. doi: 10.1038/s41568-020-00327-9. Epub 2021 Jan 29. PMID: 33514930.

A more thorough histopathological description about the data sets is missing. What exact (sub)type / stage / grade of tumors was the model trained on? Was it muscle-invasive tumors (pT2 or higher) or pTa / pT1 tumors or both? This is highly relevant, as pTa / pT1 tumors have a higher frequency of FGFR alterations and the model might just be detecting this somewhat bland morphology rather than the actual molecular alterations. How did the model perform if the cohorts were stratified by tumor stage (NMIBC vs. MIBC), grading etc.? The example provided in Figure 3 is NOT convincing. Among other things, it does not seem to show a muscle invasive tumor but rather a pTa lesion. The histomorphological criteria mentioned in the text and ref. 31 can hardly be recapitulated in the image and are not really

meaningful in routine pathology practice. The image resolution is low, the classification probability can hardly be seen and it is not clear what the color actually indicates. There is no scalebar, etc. What would an FGFR “wt / not-altered” case look like? Have these features been quantified in any way or is this just an example which fits the literature? Much more data would be needed to address this actually interesting point.

3. Results / Interpretation

While the algorithm seems to perform a little bit better than random guessing, to me the results are not convincing. This holds true in particular for the “Retrospective” and “Prospective Validation”. The model seems to be skewed towards predicting the majority of cases as FGFR+, which depending on the distribution of the cases is not really helpful, even in a pre-screening setting. This could be better evaluated, if the authors had provided Precision-Recall-Curves and F1 metrics in addition to the ROC curves. Also it confuses me that out of 24 patients (in production) there were no actual results in 5 patients (2 x insufficient tissue, 1 x test cancelled, 2 x error). This is more than 20%. How does this high dropout rate influence the proposed prescreening benefit?

Reviewer #3 (Remarks to the Author):

The study by Ramon et al. explores the potential of deep learning algorithms to serve as a pre-screening tool for identifying FGFR+ status among patients with bladder cancer. Given that FGFR+ status informs therapeutic decisions, the algorithm, if successfully deployed, could cut down on cost and unnecessary genetic testing. The study distinguishes itself from prior work by utilizing a broader range of H&E whole slide images, aggregated from three distinct datasets. The algorithm exceeded the 0.9 sensitivity threshold on three test datasets but marginally missed this benchmark in retrospective validation. The study culminates with a month-long deployment of the FGFR device in a clinical trial and prospective validation.

Comments:

Is the rate of mutations found in each of the 3 data sets the same? Is it the same as the expected population for real world evidence deployment?

Were mutation call thresholds tuned via the same 5-fold cross validations as the other algorithm hyperparameters?

Figure 2: Would it be more useful to just show each of the sub-cohorts plus the total? It unclear why this combination of cohorts was selected.

Figure 3: Figure needs scale bar. The authors should show correspondent attention-based figures here to complement the local call-levels. Pathology review should include both positive and negative instances to show what a "non-FGFR" pattern looks like from lowest-scoring tiles.

Can the authors provide more details on device cost and cost-savings? Perhaps with cost ranges for the different countries that participated for the cancelled test savings. How about cost savings estimates for a full trial as described?

If the trial is read out, can the authors provide an association between the strength of the patient-level device call (amount of FGFR-ness) and the RECIST response?

Reviewer #4 (Remarks to the Author):

Reviewer #1 (Remarks to the Author): Expert in bladder cancer targeted therapy, genomics, biomarkers, and FGFR

Ramon et al perform an analysis of H&E digital images with paired genomic data taken from a range of sources including public repositories and commercial clinical trials including FGFR inhibitors. They outline the discovery of a FGFR related morphological biomarker in a large cohort (>3000 patients) using a deep learning techniques and validate this in a separate retrospective patient cohort enriched for FGFR+ patients. The validated biomarker was then prospectively deployed in a clinical trial setting with exploratory outcomes including total analysis time of the device and influence on clinical decision to proceed to molecular testing.

This is an extremely well written manuscript which follows the biomarker road map (and medical device standards) and is logically presented. The statistical analyses seem sound and appropriate conclusions have been reached. The particular strengths are the large sample numbers as part of the discover data set and the attempt to prospectively assess in a non-interventional clinical trial. Patient selection of FGFR inhibitors for patients with urothelial cancer (and other tumours) is of key importance and this manuscript will be of interest to a wide audience including oncologist, pathologists, computer scientists and biomarker developers.

#1. This is the most thorough attempt at deploying a FGFR digital pathology algorithm and as such would strongly recommend publication in nature communications. However given that poor specificity, the main utility of the device in its current form is quickly identifying patients that are unlikely to be FGFR+ and therefore start an alternative therapy without delay. I think this will have a minimal impact on the number of genomic tests performed and overall patient management.

Response: Thank you very much for your feedback. We would first like to clarify that the algorithm was developed to the specifications of the clinical trial team with physician input (i.e., the algorithm was developed specifically to achieve high-sensitivity, at the expense of false positives, to align with the needs of the clinical study as determined by the clinical, regulatory, and operations teams); in Lines 21-23 on Page 11.

We would also like to further emphasize that the use of the device given the performance reported in the study could result in a reduction of 28.7% in genetic testing in a clinical trial setting, which could result in significant cost reductions throughout the course of a trial's lifespan. Recognizing the limitations of a low-specificity model, we also explored the potential impact of a 3-tier model. We show that the device could provide actionable clinical insight to physicians by allowing them to prioritize or de-prioritize patients for genetic testing in a standard clinical setting, where genetic testing may not be part of standard of care. Specifically, that there is potential to enrich High- to Low-likelihood groups >8x. This could help improve patient management but could also enhance the precision of care decisions, ultimately leading to better patient outcomes. To demonstrate the implications, we have added the cost-analysis in Figure 7, Lines 10-26 on Page 10, and the economic implications in Lines 32-44 on Page 13 and Lines 1-13 on Page 14.

#2. Greater insights into how the test could be further improved would be useful to the community.

Response: We have added a description of how the AI-based test could be improved in the Discussion (Lines 3-13, page 12), along with a reference to our team's recent work published at ASCO, which shows improved performance of the test in NMIBC and pan-tumor datasets after pre-training using self-supervised learning [43].

#3. The presentation of the ANNAR dataset in Figure 1 is misleading. It depicts this study as prospective validation

whereas 171 patients in the trial were used as part of the retrospective validation. Why were the data from BLC3001 and ANNAR kept separate rather than combining as a single retrospective validation set?

Response: We apologize for the confusion. We have updated Figure 1 and clarified in Lines 19-22, page 3 that we followed a stepwise validation approach, first with a well-powered Retrospective Validation (using BLC3001 data) to decide if the algorithm would be implemented in the clinical workflow, followed by a Deployment Setting Validation (using ANNAR data) to assess the workflow integration of the algorithm in the proposed deployment trial. We also added a clarification of why we did not use samples from ANNAR for the Retrospective Validation in the Discussion section, Lines 26-44, Page 12 and Lines 1-4, Page 13.

#4. Throughout the manuscript, there is mention about prospective validation of the device – eg paragraph 5 of the introduction. Given that this was insufficiently powered to show a statistically significant difference, it is unable to validate the findings. The term used in Figure 1 is more accurate as ‘a deployment trial’ and this should be made clearer earlier in the text of the manuscript.

Response: We have updated the terminology in Figures 1 and 6 and across the paper. We refer to it now as “Deployment Setting Validation” to clarify the validation was to assess workflow integration of the algorithm in the proposed deployment trial (ANNAR (NCT03955913)). We also added Lines 19-22 in Page 3 to clarify the purpose of each validation step early in the manuscript and in the Discussion section Lines 26-44, Page 12 and Lines 1-4, Page 13 to explain the reasoning behind it.

#5. The device positivity threshold has been set with a >90% sensitivity but this is at the cost of specificity which is poor around 25%. There is no mention in the discussion about what threshold is deemed clinically relevant. Notably, the simulated 3-tier model in figure 4 compares low and high device predictions and avoids the mid-risk prediction which predicts an 11.4% positivity, a similar rate of positivity to BRCA+ in prostate cancer which is now routinely performed in many countries.

Response: We agree that setting a high sensitivity threshold (90%) results in a low specificity. This high-sensitivity threshold was required by our clinical and trial operations teams for deployment of the device in the clinical study. We have added Lines 34 on Page 3 and in Lines 21-23 on Page 11.. Note that, even though the goal was to reduce trial costs, key considerations were to: 1) ensure that FGFR+ patients would not be removed from the trial given the low prevalence of FGFR in the population, 2) avoid potentially biasing the patient population entering the trial.

In the simulated 3-tier example, note that the middle tier of the 3-tier model can be treated as if not AI test was run (i.e., treat them how you would otherwise). The 3-tier model still provides actionable information on ~42% of patients. While we aim to provide actionable insights on as many patients as possible, being able to do so for 42% of patients is a significant improvement over the status quo where physicians have no means of refining their pre-test probability beyond population-level estimates (i.e., baseline prevalence).

#6. In the discussion, there is no mention about how independent the H&E FGFR device output is compared to conventional and histopathological characteristics. For example in UC, platinum sensitivity is associated with FGFR+ and in pan-tumor analysis sex, race, and histological variant has been associated with FGFR status. Were these clinical variables reviewed and could they be integrated into a model to improve the positive predictive value?

Response: We did not evaluate the outputs of the H&E FGFR device in relation to conventional histopathological characteristics such as platinum sensitivity, race, and histological variant. This was mainly because these variables were not readily available in several datasets. To elaborate, the clinical trial images used here came from the Screening step of the trial, where patients are tested for eligibility for the study. Notably, patients must be FGFR+ to be treated in the study, so the FGFR molecular test was performed during this Screening step. If patients were FGFR-, they were not treated in the study and detailed clinical data was not made available for those patients, though their informed consent did allow their pathology slides to be used for subsequent biomarker research. Thus, we cannot provide specific insights on how the device output correlates with these clinical variables. However, we have added stratified results by gender and age in supplemental Figure 3.

Although it is possible to integrate these clinical variables into a model to potentially enhance the positive predictive value of the H&E FGFR device, it should be noted that in our deployment setting, this information was not available. Therefore, the model could not be effectively deployed for the Intended Use if those clinical metrics were required as additional inputs.

#7. With a set of rules, can a trained pathologist denote FGFR+ with similar accuracy?

Response: That is a great suggestion and something we could consider in a future study. However, it is important to note that evaluating FGFR based on H&E morphology is not a routine practice in pathology and the associated morphologies referenced in previous papers are not commonly used. A review of the top predicted tiles by the algorithm in FGFR+ true positive WSIs shows that the features in these top tiles are in keeping with features previously described in a handful of FGFR+ tumors [33] (added Figure 3 and Lines 3-14, Page 6). The features however are relatively generic; pathologists might therefore show an even lower specificity than the algorithm; however the development and testing of a human-based FGFR algorithm was outside of the scope of this study. We have added this clarification in the Discussion section (Lines 14-25, Page 12).

Reviewer #2 (Remarks to the Author): Expert in deep learning for digital pathology, whole-slide imaging, and cancer therapy

In their manuscript „Development and deployment of a histopathology-based deep learning algorithm for patient pre-screening in a clinical trial for targeted-therapy in bladder cancer” Ramon et al. trained a deep learning model on image data of histopathological slide to predict FGFR-Alterations in bladder cancer. They used different training and validation cohorts originating from the TCGA-BLCA project and different pharmaceutical trials.

While the overall topic is interesting, there are several major drawbacks of the manuscript in my opinion:

#1. There are several studies which have used almost the exact same research question, similar methodology and achieved comparable results. For example ref. 33 also used the TCGA dataset to predict certain FGFR alterations from H&E slides using conv nets. The performance of the current study is comparable ref. 33 despite using substantially more slides. What is truly new and unique about the authors approach? Including two new types of alterations in the same signaling axis does not really count as novelty to me. Neither does using more slides, without substantially improving performance. Similar studies include work done by Velmahos et al., Woerl et al., and others.

- Velmahos CS, Badgeley M, Lo YC. Using deep learning to identify bladder cancers with FGFR-activating mutations from histology images. *Cancer Med.* 2021 Jul;10(14):4805-4813. doi: 10.1002/cam4.4044. Epub 2021 Jun 10. PMID: 34114376; PMCID: PMC8290253.

- Woerl AC, Eckstein M, Geiger J, Wagner DC, Daher T, Stenzel P, Fernandez A, Hartmann A, Wand M, Roth W, Foersch S. Deep Learning Predicts Molecular Subtype of Muscle-invasive Bladder Cancer from Conventional Histopathological Slides. *Eur Urol.* 2020 Aug;78(2):256-264. doi: 10.1016/j.eururo.2020.04.023. Epub 2020 Apr 27. PMID: 32354610.

- Zhang X, Wang J, Lu J, Su L, Wang C, Huang Y, Zhang X, Zhu X. Robust Prognostic Subtyping of Muscle-Invasive Bladder Cancer Revealed by Deep Learning-Based Multi-Omics Data Integration. *Front Oncol.* 2021 Aug 6;11:689626. doi: 10.3389/fonc.2021.689626. PMID: 34422643; PMCID: PMC8378227.

Response: We agree that there are several papers that tackle similar research questions. Velmahos was previously referenced and described in the Introduction section, Lines 23-26, Page 2. Woerl and Zhang papers present algorithms to predict Subtype rather than FGFR status, but we have also added them as references [31-32].

While Loeffler, et al. demonstrated a strong proof-of-concept for using CNNs to infer FGFR status from H&E-stained whole slide images, they acknowledge that analysis of larger, multi-center cohorts is needed. Here, we follow up on that initial research with an algorithm that was trained on more data, followed by a well-powered validation study, and active deployment in the clinic. We also show here that training the algorithm with more data resulted in a model that performed better than previously published results on the TCGA cohort and had strong generalizability to new data sets. Having more slides may not always translate to better performance, however, it can result in a more robust algorithm that generalized across different datasets (as we present in Figure 2) and it can provide greater confidence around the estimate of performance.

We also want to take this opportunity to emphasize that the novelty we are aiming to present in this paper is not the AI method itself, but in the robust validation and real-time deployment of the algorithm to demonstrate the potential impact this type of AI-based algorithm can have in the clinic. Most of the previous work never went beyond preliminary exploration; the algorithms were not validated on separate datasets and were not deployed in a clinical setting. Here, we show deployment of the device in a clinical study with 89 global study sites across 9 countries after a rigorous validation of the device on large datasets (following international standards for device quality management systems). Developing algorithms in a research setting is challenging, but taking the next step

and deploying them in a way that might impact clinical decisions and patients introduces a new level of complexity that are necessary to address in order to realize the impact of this technology.

Furthermore, we provide in-depth analysis and discussion of the utility and value of the device to improve trial efficiency. In the current setting, we achieved a projected reduction of 28.7% in molecular testing, and showed the potential clinical utility for prioritizing (or de-prioritizing) patients for genetic testing in a standard clinical setting, (with an enrichment of >8x for High- to Low-likelihood groups). This can be found in Lines 16-20, Page 11 and in the additional analyses we now show in Figure 7, Lines 10-26 on Page 10, and the economic implications in Lines 32-44 on Page 13 and Lines 1-13 on Page 14.

2. The algorithm which was used (Pretrained Conv Net with MIL) is somewhat outdated and there are newer methods (i. e. vision transformer) available. Furthermore, it is difficult to really assess the underlying code base as it has not been made publicly available. This is unusual. What is the reason for not providing the code and models a) for review purposes and b) as open source to the scientific community? Particularly as the model has been trained on data funded by the public and assumingly with open source methods?

Response: We agree that the methodology used here is now outdated and that newer approaches (e.g., Self-Supervised Learning, Vision Transformer, etc) have the potential to improve performance. We are currently leveraging these in our ongoing work; however, at the time this model was developed and deployed these methods were not as mature. We have added a description of how the AI-based test could be improved in the Discussion (Lines 3-13, page 12), along with a reference to work published by our team [43], showing how self-supervised methods improved performance and generalizability of the algorithm in NMIBC and pan-tumor datasets.

We would also like to clarify that the majority of the data was collected from multiple Janssen R&D clinical studies (NCT03955913, NCT03390504, NCT03473743) where data was approved for research use. This data is not publicly available due to reasons of data sensitivity, including research participant's privacy/consent. The full code base from the FGFR device is not publicly disclosed to safeguard Janssen R&D intellectual property and to ensure compliance with all relevant privacy obligations. However, we have provided a detailed technical description of the FGFR device in the online Methods (pages 15-16), together with the device structure in supplemental Figure and pseudocode in Table 1 (page 16) to facilitate the understanding of the device steps for biomarker prediction from H&E WSIs. This includes a description of what sections of the pipeline were based on open-source code, which can be downloaded at <https://github.com/CODAIT/deep-histopath/tree/master/deephistopath/wsi>.

We have included this information and added links to download the public data and/or source code in the Data and Code Availability sections of the article (Lines 18-32, page 19)

#3. Although the authors should be commended for using various test and validation data sets, I'm confused about the exact terminology. As far as I'm aware, the term "testing data" should be reserved for unseen data from an external source to make the final performance estimation. Fig. 1 somewhat indicates a different use.

Response: Thank you for your suggestion. We have updated the terminology throughout the manuscript and updated Figure 1 to be consistent and address these concerns.

#4. Furthermore, it seems to be the case that some patients from the "Development Datasets" (specifically 350 WSI from the BLC3001 trial) have been used as "Retrospective Validation". This is not standard. Why didn't the authors use the BLC2002 trial or the TCGA cases as "Retrospective Validation"? Assuming characteristics about future cases the model should be used on and tweaking the data accordingly is problematic (see Kleppe et al.).

- Kleppe A, Skrede OJ, De Raedt S, Liestøl K, Kerr DJ, Danielsen HE. Designing deep learning studies in cancer diagnostics. *Nat Rev Cancer*. 2021 Mar;21(3):199-211. doi: 10.1038/s41568-020-00327-9. Epub 2021 Jan 29. PMID: 33514930.

Response: There were no patients used in both Development Dataset and Retrospective Validation and the data split was performed prior to start of development, following the guidelines from the international standard for medical device quality management systems (ISO 13485). While planning our Validation study, we performed a power analysis and determined that 350 patients would provide adequate power to evaluate the model. We set aside 350 of the patients from BLC3001 and used the rest for model development. We added a clarification in the caption from Figure 1 and Lines 19-21 on Page 6. We also added an explanation in the Discussion, Lines 26-44 on Page 12 and Lines 1-4 on Page 13.

#4. A more thorough histopathological description about the data sets is missing. What exact (sub)type / stage / grade of tumors was the model trained on? Was it muscle-invasive tumors (pT2 or higher) or pTa / pT1 tumors or both? This is highly relevant, as pTa / pT1 tumors have a higher frequency of FGFR alterations and the model might just be detecting this somewhat bland morphology rather than the actual molecular alterations. How did the model perform if the cohorts were stratified by tumor stage (NMIBC vs. MIBC), grading etc.?

Response: Apologies for the lack of clarity. The algorithm was developed using WSIs from patients enrolled in clinical trials where one of the eligibility criteria was metastatic or locally advanced urothelial carcinoma (BLC2002/NCT03473743 here; BLC3001/NCT03390504 here). The samples tested do not have block-level staging information, however the prevalence of FGFR in our dataset is similar to the reported prevalence of FGFR in an MIBC population (12.5%, 11.6% and 15.7%, averaging ~12%) [10]. The algorithm was developed specifically to run on patients diagnosed with MIBC, hence we have not evaluated the performance of the algorithm for NMIBC patients, however, we have added a reference with our team's most recent work [43] and discussed in Lines 10-13, page 12, which shows the performance of the AI-based test in an NMIBC cohort and compares it to a newer method that leverages self-supervised training.

#5. The example provided in Figure 3 is NOT convincing. Among other things, it does not seem to show a muscle invasive tumor but rather a pTa lesion. The histomorphological criteria mentioned in the text and ref. 31 can hardly be recapitulated in the image and are not really meaningful in routine pathology practice. The image resolution is low, the classification probability can hardly be seen and it is not clear what the color actually indicates. There is no scalebar, etc. What would an FGFR "wt / not-altered" case look like? Have these features been quantified in any way or is this just an example which fits the literature? Much more data would be needed to address this actually interesting point.

Response: Thank you for your guidance. We have reformatted Figure 3 to show three low power, MIBC resection WSIs that represent some of the randomly selected true positive WSIs, and three low power, MIBC resection WSIs that represent some of the randomly selected true negative WSIs. In addition, we show the top scoring tile for each WSI and have made correlations with the few, limited histological descriptions of FGFR mutated tumors in the literature. General pathologists have not been trained to infer the presence of an FGFR alteration (or others) from an H&E-stained slide, and thus should not be considered as a point of reference on which to evaluate algorithm performance. Nevertheless, insights from a pathology standpoint can be gleaned from the algorithm's attention scores, which can highlight regions of the WSI deemed to be more (or less) informative in the algorithm's biomarker prediction. We have included this in Lines 3-14, Page 6 and Lines 14-25, Page 12 in the Discussion section.

#6. While the algorithm seems to perform a little bit better than random guessing, to me the results are not convincing. This holds true in particular for the “Retrospective” and “Prospective Validation”.

Response: Thank you for your feedback. We acknowledge that the performance is not as high as we would like for a standalone diagnostic. More recent improvements have further increased the performance of similar algorithms we are developing. However, the obtained performance is significantly higher than AUC=0.5 (random guessing) and can have a significant clinical impact at the current level of performance. Specifically, we have considered separate applications of the Device to “Rule Out” patients and to “Rule In” patients.

Rule Out approach: We would like to emphasize that the performance values obtained during Retrospective Validation (AUC=0.75 and a specificity of 31.8% at 88.7% sensitivity) indicate a projected reduction of 28.7% in genetic testing when deployed. Note that FGFR genes are only mutated in 10-20% of advanced/metastatic urothelial cancer patients [16, 17], which results in a majority of test results coming back FGFR- in clinical trials. The deployed device would minimize unnecessary molecular testing of patients that are unlikely to harbor genetic mutations, providing significant cost savings for the execution of a clinical trial. It would have the additional benefit to patients/physicians of providing actionable insight in a rapid timeframe, which could enable faster clinical decision-making and could minimize delays to treatment for some patients.

Rule In approach: While this approach was not the priority in this study, we discuss the potential impact of a High-Specificity model in which patients who are likely FGFR+ could be surfaced and prioritized for molecular testing. Given the performance of the model (see: Retrospective Validation), a High-Specificity model would result in a significantly enriched cohort (PPV approaching 50%). This results in a cohort where 1 in 2 molecular tests comes back positive, as opposed to the ~1 in 6 if patients were selected for testing at random (as would happen with the status quo where no screening tool is available).

We present some of these results in Figure 4 and we have added further analysis in Figure 7 to demonstrate the value of the device under different scenarios. Please also see: Lines 32-44, Page 13 and 1-13, Page 14 for additional discussion about the impact of an algorithm with this performance. Please, also refer to Response #1 from Reviewer #1 for further information.

#7. The model seems to be skewed towards predicting the majority of cases as FGFR+, which depending on the distribution of the cases is not really helpful, even in a pre-screening setting. This could be better evaluated, if the authors had provided Precision-Recall-Curves and F1 metrics in addition to the ROC curves.

Response: Note that the model was purposely tuned to maintain a high sensitivity (90%). Hence the majority of cases are predicted FGFR+. The high-sensitivity threshold was required by our clinical, regulatory, and trial operations teams for deployment of the device. Note that, even though the goal was to reduce genetic testing, their priority was also to ensure that FGFR+ patients would not be removed from the trial, which would slow down enrollment and have the potential to bias the patient population. We have added Lines 34 on Page 3 and Lines 21-23 on Page 11 from the Discussion section to clarify. We have also updated Figure 2 to show curves for all datasets and auPR metrics, and added supplemental Figure 3 with Precision-recall curves stratified by age and gender. Please, also refer to Response #5 from Reviewer #1 for further information.

#8. Also it confuses me that out of 24 patients (in production) there were no actual results in 5 patients (2 x insufficient tissue, 1 x test cancelled, 2 x error). This is more than 20%. How does this high dropout rate influence the proposed prescreening benefit?

Response: Thank you for your comment. Indeed, results were indeterminate for some patients.

Two WSIs could not be processed by the algorithm because the image was deemed of insufficient quality (there were insufficient remaining tiles after QC to perform a prediction). Please see Supplemental Figure 1 and pseudocode in Table 1 for details about how the device has incorporated quality checks into the workflow after an image is provided. Had we rescanned the slide or recut the block, this obstacle might have been overcome. Indeed, we view this as a positive as it prevented the reporting of a result that might have been unreliable.

One out of five FGFR- predicted patients had their molecular test cancelled by the investigator after the results of the AI device were provided, suggesting the investigator deemed the molecular test unnecessary based on the AI results. In future studies we anticipate a higher rate of cancellations as the technology becomes more established or accepted.

Two molecular tests showed insufficient tissue. As with any clinical assay, the test requires enough substrate to deliver a reliable result. Molecular tests require substantial amounts of tissue and often fail to detect the target due to poor DNA/RNA quality or low tumor purity [15]. We view this as an opportunity for the AI algorithm to provide insight into the potential FGFR status despite not having the results of the molecular test.

Reviewer #3 (Remarks to the Author):

The study by Ramon et al. explores the potential of deep learning algorithms to serve as a pre-screening tool for identifying FGFR+ status among patients with bladder cancer. Given that FGFR+ status informs therapeutic decisions, the algorithm, if successfully deployed, could cut down on cost and unnecessary genetic testing. The study distinguishes itself from prior work by utilizing a broader range of H&E whole slide images, aggregated from three distinct datasets. The algorithm exceeded the 0.9 sensitivity threshold on three test datasets but marginally missed this benchmark in retrospective validation. The study culminates with a month-long deployment of the FGFR device in a clinical trial and prospective validation.

Comments:

#1 Is the rate of mutations found in each of the 3 data sets the same? Is it the same as the expected population for real world evidence deployment?

Response: We added the rate of mutations for each dataset in Lines 14-15, Page 3. Recent studies [15,16] estimate FGFR genes are only mutated in 10-20% of advanced/metastatic urothelial cancer patients (T2 and above). Our average prevalence across datasets is ~12%.

#2 Were mutation call thresholds tuned via the same 5-fold cross validations as the other algorithm hyperparameters?

Response: We optimized hyperparameters on each cross-validation fold and based on the optimized hyperparameters we then selected the threshold providing a sensitivity above 0.9 (target minimum sensitivity required by the clinical and trial operations team).

#3 Figure 2: Would it be more useful to just show each of the sub-cohorts plus the total? It unclear why this combination of cohorts was selected.

Response: We originally decided to only show 1) the AUC on all development datasets combined (i.e., TCGA, BLC2002 and BLC3001), 2) the AUC on the dataset with closest population to the deployment setting (i.e., BLC3001 is closest to ANNAR out of the three datasets) and 3) the AUC on an independent dataset with tissues from multiple tumor sites (i.e., PAN-tumor performance and independent dataset from those used for development) to reduce the amount of lines in the ROC curve plot and allow better visualization of the main curves. Also, because the Retrospective Validation (performed under Design Controls and after Model Onboarding) was performed on BLC3001 data (350 samples), we wanted to facilitate a direct comparison between the algorithm's performance during development (i.e., Figure 2) and Retrospective Validation (i.e., Figure 4) on that dataset specifically. However, we have updated Figure 2 to show each of the performance on each of the sub-cohorts.

#4 Figure 3: Figure needs scale bar. The authors should show correspondent attention-based figures here to complement the local call-levels. Pathology review should include both positive and negative instances to show what a "non-FGFR" pattern looks like from lowest-scoring tiles.

Response: We have reformatted Figure 3 to show three representative low power MIBC resection WSIs of predicted true positives and three of predicted true negatives. In addition, we show the top scoring tile for each WSI and have described observations aligned with the few, limited histological descriptions of FGFR mutated tumors in the

literature. We have added the pathologist interpretation of these images in Lines 3-14 from Page 6.

#5 Can the authors provide more details on device cost and cost-savings? Perhaps with cost ranges for the different countries that participated for the cancelled test savings. How about cost savings estimates for a full trial as described?

Response: Thank you for the suggestion. We have added a cost-saving analysis in Figure 7, and discussed in Lines 10-26, Page 10, in Lines 30-44, Page 13 and in Lines 1-11, Page 14.

#6 If the trial is read out, can the authors provide an association between the strength of the patient-level device call (amount of FGFR-ness) and the RECIST response?

Response: Thank you for the great suggestion. We agree that this analysis could generate valuable insights to support our understanding of the disease population and drug efficacy. Unfortunately, the ANNAR trial was non-interventional (i.e., no patients were treated during the study), so the data required for these analyses is not available. However, we are planning to evaluate this in a future study leveraging data from future trials.

REVIEWER COMMENTS

Reviewer #1 (Remarks to the Author):

I have reviewed the revised manuscript and am happy with the authors efforts to address my concerns. I recommend publication.

Reviewer #2 (Remarks to the Author):

Although some of my initial major (e. g. novelty) and minor (e.g. scale bars in the images, which reviewer #3 also pointed out) concerns still stand, I feel that the manuscript has improved substantially, and the point of the study has become clearer. I would thus follow the other reviewers' opinion and endorse the manuscript.

Reviewer #2 (Remarks on code availability):

The authors have not provided the original code base but some pseudocode to show the concept. So the actual code base can not be assessed.

Reviewer #3 (Remarks to the Author):

My comments have now been appropriately addressed.

Reviewer #4 (Remarks to the Author):

Response to Manuscript Review for “Development and deployment of a histopathology-based deep learning algorithm for patient pre-screening in a clinical trial”

REVIEWER COMMENTS

Reviewer #1 (Remarks to the Author): I have reviewed the revised manuscript and am happy with the authors efforts to address my concerns. I recommend publication.

Reviewer #2:

(Remarks to the Author):

Although some of my initial major (e. g. novelty) and minor (e.g. scale bars in the images, which reviewer #3 also pointed out) concerns still stand, I feel that the manuscript has improved substantially, and the point of the study has become clearer. I would thus follow the other reviewers’ opinion and endorse the manuscript.

(Remarks on code availability):

The authors have not provided the original code base but some pseudocode to show the concept. So the actual code base cannot be assessed.

Thank you for your feedback.

We were delighted to see that all reviewers endorse the manuscript for publication. To address the availability of the codebase, we have added lines 32-36 on Page 19 of the manuscript and included a link to a repository hosted under Johnson & Johnson’s GitHub account that contains the necessary code and data to replicate the results presented in the manuscript: https://github.com/johnsonandjohnson/FGFR_Device_Review. Please note that the repository is currently private, pending final review by our legal team. However, we anticipate making it public within the next couple weeks. In the meantime, we are providing you access to the same code repository on my personal GitHub account solely for use during the review process. You can find the code here: https://github.com/albertmistu/fqfr_device_review. Please be aware that the code in this link will be removed once the version in Johnson & Johnson’s GitHub account becomes public.

Reviewer #3 (Remarks to the Author): My comments have now been appropriately addressed.

Reviewer #4 (Remarks to the Author): I co-reviewed this manuscript with one of the reviewers who provided the listed reports. This is part of the Nature Communications initiative to facilitate training in peer review and to provide appropriate recognition for Early Career Researchers who co-review manuscripts.

REVIEWERS' COMMENTS

Reviewer #2 (Remarks to the Author):

I have reviewed the code and it seems fine. Of note, as the authors point out in the revised version of the manuscript, the code can only be used to recalculate the performance metrics and generate the diagrams of the paper. It can NOT be used to recreate any of the deep learning experiments.

Reviewer #2 (Remarks on code availability):

I have reviewed the code and it seems fine. Of note, as the authors point out in the revised version of the manuscript, the code can only be used to recalculate the performance metrics and generate the diagrams of the paper. It can NOT be used to recreate any of the deep learning experiments.